# Why ReLU? A Bit-Model Dichotomy for Deep Network Training

**Ilan Doron-Arad** [1]    **Elchanan Mossel** [1]

## Abstract

Theoretical analyses of Empirical Risk Minimization (ERM) are standardly framed within the Real-RAM model of computation. In this setting, training even simple neural networks is known to be $\exists \mathbb{R}$-complete—a complexity class believed to be harder than NP, characterizing the difficulty of solving systems of polynomial inequalities over the real numbers. However, this algebraic framework diverges from the reality of digital computation with finite-precision hardware. In this work, we analyze the theoretical complexity of ERM under a realistic bit-level model ($\text{ERM}_{\text{bit}}$), where network parameters and inputs are constrained to be rational numbers with polynomially bounded bit-lengths. Under this model, we reveal a sharp dichotomy in tractability governed by the activation function: for deep networks with *any* polynomial activation with rational coefficients and degree at least 2, deciding $\text{ERM}_{\text{bit}}$ is $\#P$-hard, determining the sign of a single partial derivative is unlikely to be in BPP, and deciding a specific bit in the gradient is $\#P$-hard. In contrast, for piecewise-linear activations such as ReLU, precision requirements remain manageable: $\text{ERM}_{\text{bit}}$ is in NP, indeed NP-complete, and standard backpropagation runs in polynomial time, showing that finite-precision constraints are not merely implementation details but fundamental determinants of learnability.

## 1. Introduction

A standard theoretical formulation of empirical risk minimization (ERM) for neural networks asks the following decision question: given a rationally specified network architecture, a finite rational dataset, a (poly-time computable) loss, and a threshold $\gamma$, decide whether there exists a *real* parameter vector $\theta$ in these domains whose total loss is at most $\gamma$. In this *real-valued* ERM formulation, even for

one hidden layer ERM becomes $\exists \mathbb{R}$-complete (Bertschinger et al., 2023; Abrahamsen et al., 2021; Hankala et al., 2024): Verifying a candidate's exact solution is as hard as deciding the existence of real solutions to arbitrary systems of polynomial inequalities, so membership in NP is unknown and widely believed to be unlikely (Schaefer et al., 2024).

From the perspective of actual learning systems, this formulation hides an important issue. Modern training pipelines, and essentially all real-world implementations, operate with finite-precision arithmetic: weights, biases, inputs and outputs are stored as finite bit strings, and any realistically implementable algorithm can only manipulate such encodings (Goldberg, 1991; Higham, 2002).

To better facilitate realistic settings and aiming to bridge theory and practice, we study a natural bit-model analogue of ERM, which we call $\text{ERM}_{\text{bit}}(C)$ with parameter complexity $C = (C_1, C_2) \in \mathbb{N}^2$. An instance $I$ specifies a rationally encoded network and dataset, and the decision question is whether there exists a rational parameter vector $\theta$ for the edges such that the empirical loss $L(\theta)$ satisfies $L(\theta) \leq \gamma$, and the binary encoding length satisfies $|\text{enc}(\theta)| \leq C_1 \cdot |I|^{C_2}$. Note that every $C$ yields a different language; for simplicity, we fix a sufficiently large $C$ throughout and write $\text{ERM}_{\text{bit}} := \text{ERM}_{\text{bit}}(C)$.

Once we adopt this more concrete, bit-level formulation, the complexity of ERM shifts already for very simple architectures. When the depth is constant, and all activations and the loss are poly-time computable over $\mathbb{Q}$, a candidate parameter vector $\theta$ for $\text{ERM}_{\text{bit}}$ has a polynomial-size binary encoding, the empirical loss is a poly-time computable rational, and we can therefore *verify* $\theta$ in polynomial time, as in practice. Thus, shallow $\text{ERM}_{\text{bit}}$ lies in NP in the bit sense, unlike the standard real-numbered ERM formulation. This contrast comes entirely from how solutions are quantified, not from the underlying Turing model. This mismatch raises a basic question:

> *To what extent do hardness results for neural networks with real numbers survive when we restrict attention to polynomially encodable rational weights?*

In this paper, we study this question for *deep* networks. Deep networks are of highest practical relevance (Goodfel-

---

[1]Department of Mathematics, MIT, Cambridge, MA, USA. Correspondence to: Ilan Doron-Arad <ilanda@mit.edu>.

*Proceedings of the 43rd International Conference on Machine Learning*, Seoul, South Korea. PMLR 306, 2026. Copyright 2026 by the author(s).

low et al., 2016); however, in many theoretical aspects, they are not well understood in contrast to their shallow counterparts. We show that the algebraic structure of the activation function dictates the bit-model complexity of training: any non-linear polynomial activation functions (with rational coefficients) can encode encode computationally hard arithmetic and force extreme exploding/vanishing gradients, whereas standard piecewise-linear activations (e.g., ReLU and leaky-ReLU) admit NP verification of $\mathrm{ERM_{bit}}$ witnesses and polynomial-time back-propagation. This distinction, supported by the preference of modern deep learning architectures for piecewise linear activations, could not be given using a real-RAM setting of ERM.

## 1.1. Main Results

Our contributions can be summarized by the following theorems. First, we show that for polynomial activations, $\mathrm{ERM_{bit}}$ on deep architectures can be cast as a computationally hard numerical problem. In the next results, let id be the identity (activation) function. In the next result, we enforce the intended parameter behavior via auxiliary samples, yielding an $\mathrm{ERM_{bit}}$ instance that represents a hard numerical circuit for a wide family of polynomial activation functions. Interestingly, our next hardness results hold even for the *promise* version of the problem: Given $a < b \in \mathbb{N}$ and an instance $I$ of an optimization problem, decide whether the objective value of $I$ is at most $a$ or at least $b$.

**Theorem 1.1.** *Deciding* $\mathrm{ERM_{bit}}$ *for deep networks with activations in* $\{\sigma, \mathrm{id}\}$ *is #P-hard (under polynomial-time Turing reductions), where* $\sigma \in \mathbb{Q}[T]$ *is any non-linear polynomial activation with degree* $\geq 2$. *This result holds for* $0/1$ *loss computable in polynomial time with (i) any fixed promise gap* $a < b$, *or if (ii) all intermediate node values are uniformly bounded by a constant.*

More details on the proof are given in Section 3 and Appendix B. Recall that PH denotes the polynomial hierarchy, obtained by repeatedly adding NP-type oracle calls to polynomial-time computation. Also, $\Delta_2^P = \mathrm{P}^{\mathrm{NP}}$ is the deterministic class at the second level of this hierarchy. By the above #P-hardness, membership in NP would imply PH $\subseteq \Delta_2^P$ via Toda's theorem (Toda, 1991); hence $\mathrm{ERM_{bit}} \notin$ NP is widely believed.

**Corollary 1.2.** *If* $\mathrm{ERM_{bit}} \in$ NP, *then* PH $\subseteq \Delta_2^P$ *(in particular, the polynomial hierarchy collapses to its second level).*

This hardness does not depend on the magnitude of the numbers, as even with bounded norm in each layer, the problem remains as hard. Note that the result holds for any non-linear polynomial with rational coefficients.[1] The above result uses a non-standard (yet efficient) loss function to

obtain the #-P-hardness; however, we can obtain a slightly weaker lower bound for the standard hinge loss $\ell(x, y) = \max\{0, 1 - y \cdot x\}$ (see the discussion in Section 5 for more details).

We extend this hardness to the numerical routine most central to deep learning: backpropagation. We define BACKPROP-SIGN as the problem of determining the sign of a single gradient coordinate. Under similar assumptions to Theorem 1.1, even deciding the *sign* of a single gradient coordinate is intractable (unlikely to have a bounded-error probabilistic polynomial time algorithm (BPP)), and this persists even when all intermediate node values are uniformly bounded by a constant. Moreover, deciding the value of a specific bit in the gradient is #P-hard, making it unlikely to even be in NP.

**Theorem 1.3.** *Given a network with activation functions in* $\{\sigma, \mathrm{id}\}$, *for any non-linear polynomial activation* $\sigma$, *evaluating a single gradient coordinate at a given rational parameter* $\theta$ *admits the following hardness results.*

- *Deciding a specific bit in the gradient is #P-hard.*

- *Deciding the sign of the gradient is not in* BPP, *assuming the Constructive Univariate Radical Conjecture (Dutta et al., 2022) and assuming* NP $\not\subseteq$ BPP.

Equivalently, our reduction unconditionally shows that a polynomial-time algorithm for BACKPROP-SIGN would give a polynomial-time algorithm for POSSLP (Allender et al., 2009) (see Section 2); the constructive radical conjecture is used only to translate this into the stated BPP lower bound. We give a high level description of the proof in Section 4 and the full proof in Appendix C. The above result holds even for one-dimensional inputs and square loss with any fixed promise gap. Moreover, the hardness persists even when all intermediate node values are uniformly bounded by a constant. To the best of our knowledge, this provides the first complexity-theoretic perspective for the empirical "exploding gradients" and "vanishing gradients" phenomena (Hochreiter, 1991; 1998; Bengio et al., 1994; Glorot & Bengio, 2010; Pascanu et al., 2013). In practice, exploding gradients can be controlled by various regularization techniques. However, the above result theoretically demonstrates that, under non-linear polynomial activations, bounding the norms is insufficient.

We note that in practice "exploding/vanishing gradients" are a numerical magnitude issue rather than a computational one. Our result rules out this interpretation in the worst case: the hardness stems from a *non-benign* form of overflow/underflow, where the growth is not a simple, easily trackable scaling. Indeed, there are benign cases—e.g., repeatedly multiplying by 2 (or squaring powers of two) merely shifts the binary point / appends zeros in binary,

---

[1]if $\sigma$ is given in the input and is not universally fixed, then the intermediate values are constants that depend on $\sigma$

which can be tracked efficiently. In contrast, our constructions encode general arithmetic circuits, so intermediate quantities may have succinct descriptions but computationally hard-to-predict bit patterns.

In contrast to the above negative results, we show that standard piecewise-linear activations (such as ReLU) escape this hardness, in compliance with their abundant use in modern networks. Intuitively, unlike polynomials, they do not increase the bit-length of values multiplicatively with depth. Rather, each piecewise-linear layer performs only affine operations and comparisons, so the bit-length grows at most additively per layer, up to the bit-lengths of the rational coefficients and the fan-in.

Note that the next theorem holds even without promise, in contrast to our hardness results. This result may be known for some activations, and we give it for completeness and to highlight the contrast from polynomial activations.

**Theorem 1.4.** *For deep networks with piecewise-linear activations,* $\mathrm{ERM_{bit}}$ *is NP-complete. Furthermore, verifying a witness and computing the exact gradient (one step of back-propagation) can be done in polynomial time.*

The full proof is given in Appendix E. The above result also holds for rounding non-linear activations to a fixed number of bits.

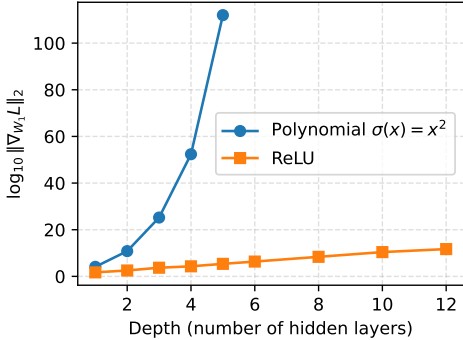

*Figure 1.* Illustration of Depth-dependent gradient growth for polynomial vs. ReLU activations. We plot $\log_{10} \|\nabla_{W_1} L\|_2$ versus depth for randomly initialized MLPs with $\sigma(x) = x^2$ or ReLU, where in both cases weights are scaled by a factor of 3 on every layer. Gradients increase with depth for both activations, but much faster for the polynomial one, qualitatively reflecting our bit-complexity separation. This experiment is illustrative rather than a formal bound.

*Remark* 1.5 (Bit-bounded activations: polynomial-time evaluation and backpropagation). Fix $k \geq 1$ with $k \leq \mathrm{poly}(|I|)$, where $|I|$ is the encoding size of the instance. Suppose every activation $\sigma^v$ is some function $\phi_k : \mathbb{Q} \to \mathbb{Q}$ that (i) is computable in time $\mathrm{poly}(|I|)$ on rational inputs, (ii) always outputs $k$-bit dyadic rationals (optionally after clipping to a fixed interval), and (iii) is equipped with a specified back-

prop rule/derivative oracle that is also computable in time $\mathrm{poly}(|I|)$ and returns rationals of bit-length $\mathrm{poly}(|I|)$. Then, one forward pass and one backward pass (computing any fixed gradient coordinate) run in polynomial time in the bit model.

This result extends to any architecture whose forward/backward computation is given by an explicit polynomial-size computation DAG in our bit model. In particular, it holds for RNN architectures run for $T \leq \mathrm{poly}(N)$ steps, by unrolling the recurrent computation into a depth-$T$ DAG and applying the same argument. Table 1 summarizes these complexity comparisons. In addition, we give

| Setting | Real-valued ERM | $\mathrm{ERM_{bit}}$ |
|---|---|---|
| Shallow ERM | $\notin$ NP[**] | NP-complete |
| Deep ERM, poly $\sigma$ | . | $\notin$ NP[*] |
| Deep ERM, pw-linear $\sigma$ | $\notin$ NP[**] | NP-complete |
| Backprop-Sign, poly $\sigma$ | in P | $\notin$ BPP[***] |
| Backprop / sign, pw-linear $\sigma$ | . | in P |

*Table 1.* Main complexity comparisons between the classical real-valued ERM formulation and the bit-bounded $\mathrm{ERM_{bit}}$ formulation. "poly $\sigma$" / "pw-linear $\sigma$" denote fixed polynomial / piecewise-linear (ReLU-type) activations; The entry $\notin$ NP[**] indicates hardness via the existential theory of the reals ($\exists \mathbb{R}$), so NP membership is unknown and believed unlikely for real-valued ERM. The entry marked [*] assumes that the polynomial hierarchy (PH) does not collapse PH $\not\subseteq \Delta_2^P$, and [***] assumes the constructive univariate radical conjecture and NP $\not\subseteq$ BPP.

an intuitive visualization of the growth rate difference between non-linear polynomials and piecewise linear activations in Figure 1. Our result offers a complexity-theoretic perspective that may help explain, at an abstract level, why piecewise-linear activations are attractive in modern deep learning architectures.

## 1.2. Related Work

**Real-RAM computation and $\exists\mathbb{R}$-hardness.** The Blum–Shub–Smale (BSS) model (Blum et al., 1989) and Smale's program on complexity over $\mathbb{R}$ (Smale, 1998) provide the standard real-RAM abstraction allowing each basic operation on real numbers to take $O(1)$. Many geometric and algebraic decision problems can be expressed in the *existential theory of the reals* ($\exists\mathbb{R}$), which asks whether a given system of polynomial equalities and inequalities has a real solution (Schaefer, 2009). A problem is called $\exists\mathbb{R}$-*hard* if it is at least as hard as any problem in this class. It is known that NP $\subseteq \exists\mathbb{R} \subseteq$ PSPACE and widely assumed that both containments are proper (Canny, 1988; Schnider & Weber, 2023; Schaefer et al., 2024). Unlike NP, an $\exists\mathbb{R}$ witness is a tuple of *real* numbers, which need not admit a succinct rational encoding; this is one reason $\exists\mathbb{R}$ is believed to strictly contain NP (Schaefer, 2009; Schaefer et al., 2024). Hence, classifying a problem

as $\exists\mathbb{R}$-complete is considered a stronger result compared to merely NP-completeness. In particular, in the standard ERM framework allowing infinite precision, training even fairly simple neural networks with one hidden layer is $\exists\mathbb{R}$-complete (Abrahamsen et al., 2021; Bertschinger et al., 2023; Hankala et al., 2024), where the completeness part assumes the real-RAM model. Thus, the real-RAM $\exists\mathbb{R}$-hardness and our bit-model #P-hardness reflect different obstructions: the former concerns the existence of exact real solutions, whereas the latter concerns computing bits of succinctly represented arithmetic computations.

**Bit-model hardness and SLP.** For our hardness results, we rely on the POSSLP and BITSLP problems (see (Allender et al., 2009; Koiran & Perifel, 2011; Bläser et al., 2024)), the problem of deciding the sign or a specific bit of a deep arithmetic circuit over the operations $\{\times, +, -\}$ and constants $0, 1$. Bürgisser and Jindal (Bürgisser & Jindal, 2024) provided conditional lower bounds for POSSLP under the constructive radical conjecture of (Dutta et al., 2022). We build on these works to obtain hardness for deep-network ERM$_{\text{bit}}$ and for BACKPROP-SIGN with polynomial activations.

**Hardness and algorithms for ERM.** The NP-hardness of training small or shallow networks in the bit model has been known since the works of (Judd, 1988; Blum & Rivest, 1988; Šíma, 2002). For ReLU and other piecewise-linear activations, (Dey et al., 2020) analyzed one-node ReLU training, (Goel et al., 2021) established sharp hardness thresholds for depth-2 networks, and (Froese et al., 2022; Froese & Hertrich, 2023) proved parameterized and fixed-dimension hardness results; (Boob et al., 2022) obtained further NP-hardness and approximation results.

Several algorithms were also given to ERM. Important examples include the works of (Arora et al., 2018; Pilanci & Ergen, 2020; Ergen & Pilanci, 2021a;b), who studied ReLU networks and derived polynomial-time algorithms under structural restrictions (e.g., fixed input dimension or width). For linear-threshold activations, (Khalife et al., 2024) give algorithms polynomial in the dataset size but exponential in network size and ambient dimension.

The recent work of (Doron-Arad, 2025) provides a lower bound for *discrete* ERM, where each parameter weight is chosen from a discrete finite set given in the input explicitly. For this model, the above paper shows that this discrete variant of ERM is computationally hard. While the setting bears similarities to our first result, the above model is much more restrictive than our general bit-level ERM. More importantly, the activation functions used by (Doron-Arad, 2025) are highly non-continuous and specifically engineered, in sharp contrast to our general family containing any non-linear polynomial of rational coefficients.

**Exploding/vanishing gradients.** Exploding and vanishing gradients have been studied extensively in recurrent and deep networks (Hochreiter, 1991; 1998; Bengio et al., 1994; Glorot & Bengio, 2010; Pascanu et al., 2013), and more recent work analyzes instabilities in deep training dynamics (Sun et al., 2022) and surveys these issues (Goodfellow et al., 2016). Our contributions are orthogonal: we construct explicit families of networks with polynomial activations whose gradients tend to be extremely large or small in both bit-complexity and magnitude sense, and we show that even deciding their sign is intractable under standard complexity assumptions. In terms of computational complexity, the work of (Backurs et al., 2017) provides a fine-grained (polynomial) lower bound for computing the gradient.

**Organization.** Section 2 formalizes our bit model, ERM$_{\text{bit}}$, and the straight-line program (SLP) primitives. Section 3 gives the proof roadmap for the #P-hardness of ERM$_{\text{bit}}$ with polynomial activations, with the full proof deferred to Appendix B. Section 4 gives the corresponding roadmap for the hardness of backpropagation, with the full proof deferred to Appendix C. The piecewise-linear tractability result is proved in Appendix E. Section 5 discusses implications, limitations, and open questions. We give in Section 6 a short conclusion.

## 2. Preliminaries

**Notations** Let $\mathbf{1}_B$ be the indicator function for some boolean condition $B$, which is equal to $1$ if $B$ is true and is set to $0$ otherwise; for some $k \in \mathbb{N}$, let $[k] = \{1, \ldots, k\}$.

**Bit Model and Rational Encoding** We work in the standard Turing machine (bit) model, where generally numbers are represented in binary and running time is measured in bit operations. A rational $q \in \mathbb{Q}$ is encoded as a reduced pair $(\text{num}(q), \text{den}(q)) \in \mathbb{Z} \times \mathbb{N}$ with $\text{den}(q) > 0$ and $\gcd(\text{num}(q), \text{den}(q)) = 1$, and its size is the total bit-length of $\text{num}(q)$ and $\text{den}(q)$. We write $|I|$ for the total bit-length of a standard encoding of an input instance $I$, and say an algorithm runs in polynomial time if it uses $|I|^{O(1)}$ bit operations. Arithmetic over $\mathbb{Q}$ is exact (addition, multiplication, and polynomial evaluation), with cost measured by bit length; unlike the real-RAM model, operations are not unit-cost. Thus, although the input instance and candidate parameters have polynomial-size encodings, intermediate rational values produced during forward and backward computation are not a priori bit-bounded; their bit-length is part of the computational cost and may grow with depth.

**Empirical Risk Minimization** We give below the formal definition of ERM in the bit model.

**Input.** An ERM instance specifies a feedforward network and a finite dataset. The network is a directed acyclic graph $N = (V, E)$ with vertex set $V = S \cup H \cup T$, where $S$ are the unique input vertices, $H$ is the set of hidden neurons, and $T$ is the set of output neurons. The *depth* is the length of the longest directed path from some $s \in S$ to a vertex in $T$. The dataset is $\mathcal{D} = \{(x_i, y_i)\}_{i=1}^n$, with inputs $x_i \in \mathcal{X} \subseteq \mathbb{Q}^V$ and labels $y_i \in \mathcal{Y} \subseteq \mathbb{Q}^{|T|}$. For the reductions, we allow inputs to have coordinates at all vertices, injected additively into the corresponding preactivations. The usual source-only model is the special case where all non-source coordinates are zero. Each $v \in H \cup T$ has a fixed activation $\sigma^v : \mathbb{Q} \to \mathbb{Q}$, and we are given a loss function $\mathcal{L} : \mathcal{Y} \times \mathcal{Y} \to \mathbb{Q}$. All $\sigma^v$ and $\mathcal{L}$ are polynomial-time computable on rational inputs.

Let $\Theta = \mathbb{Q}^E \times \mathbb{Q}^E$ be the parameter domain of the instance, which enables selecting scalar weight and bias per edge. All numerical data in the instance are rationals encoded as specified above; the *size* of the instance is the total bit-length of this encoding.

**Network semantics.** A parameter vector is a tuple

$$\theta = (w_{u,v}, b_{u,v})_{(u,v) \in E} \in \Theta, {}^{2}$$

Given $\theta$ and an input $x \in \mathbb{Q}^V$, for $s \in S$ which is a source in the network, set $f_\theta^s(x) := x_s$. For each other $v \in V$ define

$$f_\theta^v(x) := \sigma^v \left( x_v + \sum_{(u,v) \in E} \left( w_{u,v} \cdot f_\theta^u(x) + b_{u,v} \right) \right).$$

Define the network output as $f_\theta(x) := (f_\theta^t(x))_{t \in T} \in \mathcal{Y}$.

**Objective and decision problem.** The empirical loss of the parameter set $\theta$ on $\mathcal{D}$ is defined by

$$L(\theta) := \sum_{i=1}^n \mathcal{L}(f_\theta(x_i), y_i).$$

For some $C = (C_1, C_2) \in \mathbb{N}^2$, in the bit-model $\mathsf{ERM}_{\mathrm{bit}}(C)$ decision problem we consider the language

$$\mathsf{ERM}_{\mathrm{bit}}(C) = \{ I \mid \exists \theta \in \Theta \text{ s.t. } L(\theta) \leq \gamma,$$
$$|\mathrm{enc}(\theta)| \leq C_1 \cdot |I|^{C_2} \}.$$

where $I$ is an encoded ERM instance with binary bit-length $|I|$, and $\mathrm{enc}(\theta)$ is the binary encoding of $\theta$ (the certificate) with encoding size $|\mathrm{enc}(\theta)|$. Recall that we use $\mathsf{ERM}_{\mathrm{bit}} = \mathsf{ERM}_{\mathrm{bit}}(C)$ for a fixed and sufficiently large parameter complexity $C$ throughout the paper.

---

[2] Our lower bounds yield similar results if a bias is defined for every node instead for every edge.

In other words, given an instance $I = (N, \mathcal{D}, \mathcal{L}, \gamma)$, the bit-model ERM decision problem asks whether there exists a *rational* parameter vector $\theta$ whose binary encoding has length polynomial in $|I|$ such that $L(\theta) \leq \gamma$. This excludes solutions with super-polynomial encoding, which are allowed in real-RAM.

**Definition 2.1** (($a, b$)-promise $\mathsf{ERM}_{\mathrm{bit}}$)**.** Fix $a < b$ in $\mathbb{Q}$. In the ($a, b$)-*promise* version, the input is an $\mathsf{ERM}_{\mathrm{bit}}$ instance $I$ (with the same encoding conventions) and it is promised that either (i) there exists a feasible $\theta$ with $L(\theta) \leq a$ (YES case), or (ii) for every feasible $\theta$ we have $L(\theta) \geq b$ (NO case). The goal is to distinguish YES from NO.

**Complexity Classes.** The class P consists of all problems that can be solved in polynomial time by such a deterministic Turing machine. The class NP consists of all problems for which a proposed solution can be verified in polynomial time by a deterministic Turing machine, given the input and the solution. *Bounded-error probabilistic polynomial time (BPP)* describes decision problems decidable by a probabilistic Turing machine with success probability at least $\frac{2}{3}$ in polynomial time. The class #P (Valiant, 1979) is the counting analogue of NP: it contains problems that ask for the number of polynomial-length certificates (witnesses) that make a polynomial-time verifier accept a given input. While $P \subseteq BPP$ and $P \subseteq NP$, it is only conjectured that $P = BPP$ and $P \neq NP$.

**Straight Line Programs (SLP)** The paper will use *straight-line programs (SLP)* in some of the reductions.

**Definition 2.2.** A (numeric) *straight-line program (SLP) P* is a sequence of rationals $(a_0, a_1, \ldots, a_\ell)$ such that $a_0 \in \mathbb{Q}$ and $a_i = a_j \oplus a_k$ for all $1 \leq i \leq \ell$, where $\oplus \in \{+, -, *\}$ and $j, k < i$.

The definition of SLP given above is a softened definition defined for our reduction needs. Specifically, SLPs are commonly defined as univariate polynomials with $a_0 = 1$. In our lower bounds, we use either $a_0 = 1$ or $a_0 = 2^{-\ell}$, where the latter yields programs with all intermediate values in $[-1, 1]$. A thorough discussion and related results proving why the above definition is without loss of generality in our setting, and further background on the subject are provided in Appendix A.

In SLPs, we are given only the inductive representation of their output (i.e., the last variable). Importantly, we do not have an explicit encoding of the variable's values.

This leads to the following problems originating in (Allender et al., 2009) and also appears in (Koiran & Perifel, 2011; Bläser et al., 2024).

**Problem 2.3.** $\mathsf{BitSLP}$: *Given an integer $j \in \mathbb{N}$ and an* SLP *$P = (a_0, \ldots, a_\ell)$ computing a rational $u/v = a_\ell = n_P$*

with $u \in \mathbb{Z}$ and $v \in \mathbb{N}$, *decide if the $j$-th* LSB *binary digit of $n_P$ is* 1, *i.e., decide whether* $\lfloor \frac{|u|}{v} / 2^j \rfloor \bmod 2 = 1$.

A strong hardness result for this problem follows from (Allender et al., 2009).

**Theorem 2.4.** *(Allender et al., 2009):* BitSLP *is* #P-*hard*.

We also use the following slightly easier version, concerning the sign of $n_P$ rather than a specific bit, as a source of hardness for our reductions.

**Problem 2.5.** PosSLP: *Given an* SLP $P = (a_0, \ldots, a_\ell)$ *computing a number $n_P = a_\ell$, decide if $n_P > 0$.*

We will use a lower bound on PosSLP obtained by (Bürgisser & Jindal, 2024), stating that a conjecture of (Dutta et al., 2022) (see Appendix A) and a randomized polynomial time algorithm for PosSLP would imply the very unlikely statement that NP $\subseteq$ BPP.

**Theorem 2.6.** *(Bürgisser & Jindal, 2024): If* Conjecture A.2 *is true and* PosSLP $\in$ BPP*, then* NP $\subseteq$ BPP*.*

# 3. Computational Complexity of Deep Networks

In this section, we give the high level ideas in the proof of Theorem 1.1. The full proof is given in Appendix B. We start with a short road map of the proof.

**Proof roadmap.** The hardness part of Theorem 1.1 is proved by reducing BitSLP to ERM$_{bit}$. The main ingredient is a multiplication gadget: for every fixed non-linear polynomial activation $\sigma \in \mathbb{Q}[T]$, Lemma 3.1 expresses $xy$ as a fixed rational linear combination of shifted evaluations of $\sigma$. Thus, a deep network with activations in $\{\sigma, \mathrm{id}\}$ can simulate the $+, -, \times$ gates of an SLP. We then add auxiliary samples that force every low-loss solution to implement the distinguished SLP computation on the main input, and use a final bit-extraction loss so that the ERM instance separates according to the required bit of the SLP output. Below we make this more precise.

We use the following result in the construction. Let $\mathbb{F}[T]$ denote the ring of polynomials in the indeterminate $T$ with coefficients in a field $\mathbb{F}$.

**Lemma 3.1.** *Let $f(T) = \sum_{k=0}^{\mu} r_k T^k \in \mathbb{Q}[T]$, where $r_\mu \neq 0, \mu \geq 2$, be a non-linear polynomial with rational coefficients. Then, there is a polynomial-time algorithm that, given $f$, returns rational numbers $\lambda_0, \ldots, \lambda_\mu$ such that for every $x, y \in \mathbb{R}$ it holds that*

$$\sum_{j=0}^{\mu} \lambda_j \big( f(x+y+j) - f(x+j) - f(y+j) \big) = x \cdot y.$$

For example, if $f(x) = x^2$, then $\lambda_0 = \frac{1}{2}$ and $\lambda_1, \lambda_2 = 0$ satisfy the conditions of the lemma since $\frac{1}{2}((x+y)^2 - x^2 - y^2) = x \cdot y$. All coefficients $r_k$ of $f$ and all coefficients $\lambda_j$ are in $\mathbb{Q}$ and are not restricted to be integers. Note also that "polynomial" here means with integer exponents. Allowing non-integer exponents (such as $T^{1/2}$) would break the argument. We show in Appendix F that the statement cannot be generalized for rational non-integer exponents and for integer negative exponents.

We give the following reduction from BitSLP to ERM$_{bit}$, to be used in the proof of Theorem 1.1. We give in Figure 2 a road map for our SLP reductions and illustrate the construction in Figure 3.

**Lemma 3.2.** *Let $\sigma \in \mathbb{Q}[T]$ be a non-linear polynomial and let $a, b \in \mathbb{N}$ such that $a < b$. There is a polynomial-time procedure that, given a BitSLP instance $(j, P = (a_0, \ldots, a_\ell))$, constructs an ERM$_{bit}$ instance $I(P)$ with promise gap $a < b$ and activations in $\{\sigma, \mathrm{id}\}$ such that there is $\theta^*(P) \in \Theta$ satisfying $f_{\theta^*(P)}(a_0) = a_\ell =: n_P$. In addition, $P$ is a* YES*-instance of* BitSLP *if and only if $I(P)$ is a* YES*-instance of* ERM$_{bit}$ *(with a promise $a < b$). The size of $I(P)$ is polynomial in the length of $P$.*

Using the above lemma, the proof of Theorem 1.1 follows.

# 4. Computational Complexity of Gradient Descent

In this section, we show that it is unlikely that there is a randomized polynomial-time algorithm that, given a rationally specified network with polynomial activation, a dataset, a loss function, and a parameter vector, decides the sign of a single gradient coordinate.

We work in the same bit model and with the same network and loss conventions as before, but we restrict to scalar-input, scalar-output networks and the activation family introduced in the previous section. Concretely, we assume: (i) the input space is $\mathcal{X} \subseteq \mathbb{Q}$ and the label space is $\mathcal{Y} \subseteq \mathbb{Q}$; (ii) for every non-input neuron $v$ the activation is either $\sigma^v = \sigma$ or $\sigma^v = \mathrm{id}$, where $\sigma(T) = \sum_{k=0}^{\mu} r_k T^k \in \mathbb{Q}[T]$ is a fixed non-linear polynomial with rational coefficients; (iii) the loss $\mathcal{L} : \mathbb{Q} \times \mathbb{Q} \to \mathbb{Q}$ is polynomial-time computable and differentiable in its first argument, and the partial derivative with respect to the first argument is polynomial-time computable and rational-valued on rational inputs.

We regard the loss $L(\theta) = \sum_i \mathcal{L}(f_\theta(x_i), y_i)$, which is defined by an algebraic expression in the parameters, as the restriction to $\mathbb{Q}^p$ of a real-analytic function $L : \mathbb{R}^p \to \mathbb{R}$. For any scalar parameter coordinate $w_e$, we write $\partial L / \partial w_e(\theta)$ for the usual real partial derivative of this extension evaluated at the given rational point $\theta$. The Backprop-Sign problem asks for the sign of a single gradient coordinate at a

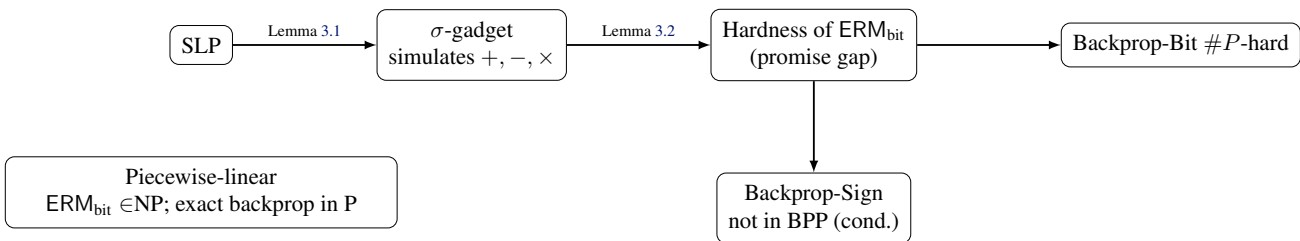

*Figure 2.* Proof roadmap: SLP based hardness is embedded into deep networks via a fixed polynomial activation $\sigma$, yielding conditional lower bounds for $\mathrm{ERM_{bit}}$ and Backprop-Sign; piecewise-linear activations admit NP verification and polynomial-time exact backprop in the bit model.

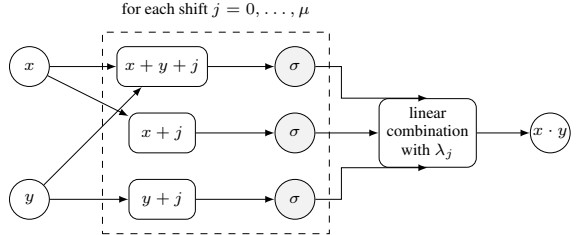

*Figure 3.* Multiplication gadget using a fixed polynomial activation $\sigma \in \mathbb{Q}[T]$. For each integer shift $j = 0, \ldots, \mu$, the network forms three affine combinations $x + y + j$, $x + j$, and $y + j$, applies $\sigma$, and then takes a fixed linear combination with coefficients $\lambda_j \in \mathbb{Q}$ (indicated by the "linear combination" node).

given rational parameter point. The Backprop-Bit problem asks for the value of the $j$-th lower bit in this gradient. We provide a concise definition of both problems below.

**Definition 4.1** (Backpropagation sign and Backpropagation bit problems)**.** An instance of Backprop-Sign with polynomial activation is a tuple $(N, \mathcal{D}, \mathcal{L}, \theta, e^\star)$ (respectively, for Backprop-Bit an instance is $(N, \mathcal{D}, \mathcal{L}, \theta, e^\star, j)$ with $j \in \mathbb{N}$) where $N = (V, E)$ is a network as above with activations $\sigma^v \in \{\sigma, \mathrm{id}\}$ for all non-input vertices $v \in V$ where $\sigma \in \mathbb{Q}[T]$ is a non-linear polynomial activation with degree $\mu \geq 2$, $\mathcal{D} = \{(x_i, y_i)\}_{i=1}^n \subseteq \mathbb{Q} \times \mathbb{Q}$ is a dataset, $\mathcal{L} : \mathbb{Q} \times \mathbb{Q} \to \mathbb{Q}$ is a polynomial-time computable loss, $\theta = (w_e, b_e)_{e \in E}$ is a parameter vector with all $w_e, b_e \in \mathbb{Q}$, and $e^\star \in E$ is a distinguished edge. The question is whether $\frac{\partial L}{\partial w_{e^\star}}(\theta) > 0$, (whether the $j$-th bit in $\frac{\partial L}{\partial w_{e^\star}}(\theta)$ is 1) where $L(\theta) = \sum_{i=1}^n \mathcal{L}(f_\theta(x_i), y_i)$ is the empirical loss. The input size is the total bit-length of all rationals specifying $N, \mathcal{D}, \mathcal{L}$ (as a program), $\theta$, and $e^\star$.

The above formulation for Backprop-Sign can be made computationally easier with a promise gap $a < b$, where we have to decide if the partial derivative is larger than $b$ or smaller than $a$. Formally,

**Definition 4.2** (Backprop-Sign with a promise)**.** An instance of Backprop-Sign with a promise gap is a tuple $(N, \mathcal{D}, \mathcal{L}, \theta, e^\star, b)$, where $(N, \mathcal{D}, \mathcal{L}, \theta, e^\star)$ is a Backprop-Sign instance and $b \in \mathbb{N}$. The goal is to distinguish between $\frac{\partial L}{\partial w_{e^\star}}(\theta) \geq b$ (the YES case) and $\frac{\partial L}{\partial w_{e^\star}}(\theta) \leq -b$ (the NO

case). Moreover, $b \leq \mathrm{poly}(|I|)$ for a fixed polynomial.

We continue with a high level description of the construction. The full proof is given in Appendix C.

**Proof roadmap.** The backpropagation hardness uses the same SLP-simulating network from Section 3, but places the hard numerical quantity inside one gradient coordinate rather than inside the ERM objective value. We choose a simple differentiable loss, a rational parameter point $\theta^*$, and a distinguished edge $e^\star$ so that $\partial L / \partial w_{e^\star}(\theta^*)$ is equal, up to a fixed rational factor, to the SLP output $n_P$. Therefore, deciding the sign of this gradient coordinate solves PosSLP, while deciding a specified bit of it solves Bit-SLP. The former gives the conditional Backprop-Sign lower bound, and the latter gives the unconditional #P-hardness for Backprop-Bit. The full details are given in Appendix C.

## 5. Discussion

This section discusses the implications and limitations of our results and outlines several open questions.

**Real vs. bit ERM.** Our results show that the choice of computational model is central to understanding the complexity of deep learning. Namely, unrealistic real numbers render the complexity of ERM far from ERM on numbers that can be represented in practice, as shown in Table 1. We believe that our $\mathrm{ERM_{bit}}$ is closer to ERM in practice and consider deriving further bounds in this model as an interesting theoretical direction with practical significance. It is worth noting that the $\mathrm{ERM_{bit}}$ model is still somewhat far from practice and should still be considered as an intermediate model that is closer to practice. For example, our positive results for piecewise linear activations for $\mathrm{ERM_{bit}}$ and Backprop-Sign overlook the empirical fact that gradient exploding can still occur in such activations (Goodfellow et al., 2016); however, such gradient explosions are incomparably small than those produced by non-linear polynomials.

**Depth, activation functions, and practical relevance.** For deep networks, we obtain a sharp contrast between

polynomial activations and standard piecewise-linear activations such as ReLU, highlighting depth and the activation as key drivers of computational hardness in the bit model. Our lower bounds apply for *any* non-linear polynomial with rational coefficients. Polynomial activations, especially quadratic ones, are well studied both in practice and theoretically (Soltani & Hegde, 2019; Allen-Zhu & Li, 2023; Blondel et al., 2017; Livni et al., 2014; Sarao Mannelli et al., 2020; Bartan & Pilanci, 2021; Du & Lee, 2018; Soltanolkotabi et al., 2019). In addition, due to Taylor expansion, used to approximate smooth activations (sigmoid, tanh, etc) (Roheda et al., 2024), polynomial activations admit strong practical motivation. Hence, even though our results are worst-case and theoretical, we believe our dichotomy gives a valuable distinction between the two families (piecewise linear and polynomial). Thus, polynomial activations are both practically motivated and serve as a clean algebraic contrast to piecewise-linear activations.

Beyond the polynomial versus piecewise-linear regimes, it is natural to ask where other nonlinearities used in practice fit into this picture, such as smooth activations or non-Lipschitz activations like $1/x$. Our current techniques do not classify these activations, and accomplishing this remains an open question. In particular, we show in Appendix F that our techniques cannot be generalized to power functions with non integer rationals or negative integer exponents.

**Exploding/vanishing gradients and "non-benign" overflow.** Our lower bounds should not be read as claiming that practical exploding or vanishing gradients are *typically* computationally hard. Rather, they show that in finite-precision models there exist worst-case instances where overflow (or underflow) is *algorithmically* non-benign: the issue is not merely that numbers become large, but that their binary representations encode hard arithmetic; thus, "tracking the magnitude" is insufficient. This contrasts with benign overflow patterns (e.g., repeated exponentiation with base 2) that correspond to simple bit shifts and are easier to predict and control. A simple depth effect already appears for $\sigma(x) = x^2$: a one-dimensional depth-$L$ network that repeatedly applies this activation maps $x$ to $x^{2^L}$. Hence inputs with $x > 1$ can explode, while inputs with $0 < x < 1$ vanish, at a rate exponential in the depth; our hardness results show that in the worst case this growth can also encode computationally hard bit patterns.

**Loss Functions** The $\#$-P-hardness result presented in Theorem 1.1 uses a non-standard loss function. This loss function is used to efficiently extract a specific bit $j$ from a very large number; this cannot be done using standard losses such as squared, hinge, etc., due to their intrinsic algebraic properties. Even though, we still can obtain a slightly weaker lower bound using the standard hinge loss

$\ell(x, y) = \max\{0, 1 - y \cdot x\}$ by considering the sign (rather than a specific bit) of the output of an arithmetic circuit. We give the details in Appendix B.

**Dimensions and Regularization** The main hardness result in Theorem 1.1 uses large input/output dimensions. A natural open question is whether the same $\#P$-hardness can be shown while keeping both dimensions $O(1)$. We note that if one allows a centered quadratic regularizer $\lambda \|\theta - \theta^*\|_2^2$, then such a constant-dimensional variant is possible as we show in the appendix (see Remark B.2). Thus, the unregularized hardness result uses polynomially large input/output dimensions, while the centered quadratic regularizer is only used to obtain a cleaner constant-dimensional variant; we do not claim such a constant-dimensional result for standard weight decay.

**Number representation of deep networks in practice.** We do not aim to model a particular floating-point format or per-operation rounding rule; rather, ERM$_{\text{bit}}$ is a clean finite-precision abstraction in which representation length is part of the computational cost. Real-world deep learning does not operate over exact reals: parameters, activations, and gradients are stored in finite-precision floating-point formats (typically IEEE 754), with rounding, a finite exponent range, and special values such as $\pm\infty$ and NaN; overflow and underflow are therefore intrinsic to the arithmetic (IEEE, 2019). Moreover, modern training commonly uses *mixed precision*: many tensors are kept in low precision (e.g., FP16/BF16/TF32) for throughput, while critical accumulations (e.g., master weights and optimizer updates) are performed in FP32; in FP16, (dynamic) loss scaling is used to mitigate gradient underflow (Micikevicius et al., 2017; Kalamkar et al., 2019; NVIDIA, 2020). Our model abstracts these representation choices while still respecting the finite-precision regime, which is invisible in real-RAM abstractions.

**Smooth activations in practice.** Smooth nonlinearities (sigmoid, tanh, softmax, GELU) are typically implemented via finite-precision routines (e.g., polynomial/rational approximations or lookup tables) followed by rounding (and often clipping or numerical stabilization). In our terminology, such implementations correspond to bit-bounded activation maps $\phi_k$ with $k$-bit outputs. Thus, for any $k \leq \text{poly}(N)$, Remark 1.5 implies polynomial-time forward evaluation and one backward pass for the resulting computation DAG. This remark concerns evaluation of a fixed finite-precision implementation, and does not assert tractability for the corresponding exact real-valued activation.

**Real-RAM vs. bit model in other domains** Our results give evidence that learning theory models are occasionally

far away from practice. Studying core learning theory questions on more realistic models, thus bridging theory and practice, is a core challenge. For example, we give in Appendix D an easy construction showing that in the setting of PAC learning, the question of how many samples are needed to learn a simple class in the realizable setting, depends on whether one works in an ideal real-number model or in a finite-precision bit model.

**Theorem 5.1.** *There exists a hypothesis class and distribution where the realizable PAC sample complexity is $O(1)$ in the exact* real-RAM *model, but scales as $\Omega(q)$ in the bit model with $q$-bit precision rounding.*

**Quantization and finite-precision training.** Our $\mathrm{ERM}_{\mathrm{bit}}$ model can be viewed as a finite-representation analogue of the standard ERM theoretical problem on real numbers. In practice, there are several reasons to perform quantization *during* training: (i) memory and bandwidth constraints on accelerators, (ii) the need to simulate the numerical behavior of low-precision deployment hardware, and (iii) empirical evidence that quantization-aware training can improve robustness and generalization for quantized models (Courbariaux et al., 2015; Rastegari et al., 2016; Polino et al., 2018; Rokh et al., 2023). Our bit-model results thus provide worst-case complexity guarantees for a clean finite-precision abstraction related to such regimes (Tailor et al., 2020; Zhou et al., 2018; Nagel et al., 2022; Sakr et al., 2022; Liu et al., 2023), rather than to a post hoc "train-then-quantize" pipeline.

## 6. Conclusion

We studied ERM in a finite-precision bit model, where candidate parameters are required to have polynomial-size rational encodings. This contrasts with the classical real-valued formulation, where even shallow ReLU networks are already $\exists\mathbb{R}$-complete, and hence membership in NP is not expected. In our bit model, constant-depth networks have polynomially checkable rational solutions, while depth together with polynomial activations introduces a new source of difficulty. For any non-linear polynomial activation with rational coefficients, deep networks can encode hard arithmetic, making $\mathrm{ERM}_{\mathrm{bit}}$ and basic backpropagation decision problems computationally intractable in the worst case. In contrast, for piecewise-linear activations such as ReLU, candidate solutions remain polynomially checkable and exact backpropagation can be performed in polynomial time. These results show that finite precision is not merely an implementation detail, but a central part of the computational model. More broadly, they suggest that studying learning problems in realistic bit models can reveal distinctions that are hidden by classical real-valued formulations.

## Acknowledgements

We thank the ICML reviewers and area chairs for helpful comments that improved the presentation. Ilan Doron-Arad is supported by NSF DMS-2031883 and the Vannevar Bush Faculty Fellowship ONR-N00014-20-1-2826 (PI Mossel). Elchanan Mossel is partially supported by NSF DMS-2031883, the Vannevar Bush Faculty Fellowship ONR-N00014-20-1-2826, and a Simons Investigator Award.

## Impact Statement

This paper presents work whose goal is to advance the field of Machine Learning. There are many potential societal consequences of our work, none of which we feel must be specifically highlighted here.

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

# A. Straight Line Programs (SLP)

In this section, we give the missing details on straight line programs (SLP). For the convenience of reading, we repeat some definitions and results given in Section 2. We start with a more general definition of SLP.

**Definition A.1.** A *straight-line program (SLP)* $P$ is a sequence of univariate integer polynomials $(a_0, a_1, \ldots, a_\ell)$ such that $a_0 = 1$, $a_1 = x$ and $a_i = a_j \oplus a_k$ for all $2 \leq i \leq \ell$, where $\oplus \in \{+, -, *\}$ and $j, k < i$. We use $\tau(f)$ to denote the minimum length of an SLP that computes a univariate polynomial $f$.

In our hardness results, we use a conjecture proposed in (Dutta et al., 2022). A *radical* $\mathsf{rad}(f)$ of a non-zero integer polynomial $f \in \mathbb{Z}[x_1, \ldots, x_n]$ is the product of the irreducible integer polynomials dividing $f$.

**Conjecture A.2.** Constructive univariate radical conjecture: *For any polynomial $f \in \mathbb{Z}[x]$, we have $\tau(\mathsf{rad}(f)) \leq \mathsf{poly}(\tau(f))$. Moreover, there is a randomized polynomial time algorithm which, given an SLP of size $\ell$ computing a polynomial $f$, constructs an SLP for $rad(f)$ of size $\mathsf{poly}(\ell)$ with success probability at least $1 - \frac{1}{\Omega(\ell^{1+\varepsilon})}$ for some $\varepsilon > 0$.*

In addition to polynomial SLPs as in Definition A.1, in this paper we use SLPs that compute numbers (rather than univariate polynomials): a *numeric* SLP is a sequence $(b_0, \ldots, b_m) \subseteq \mathbb{Q}$ such that $b_0$ is a fixed rational constant and for every $1 \leq i \leq m$, $b_i = b_j \oplus b_k$ for some $j, k < i$ and $\oplus \in \{+, -, \times\}$. For $b_0 = 1$, numeric SLP is equivalent to polynomial SLP evaluated at $x = 1$. Below, we show that we can generalize the initial constant to be a small dyadic rational, inducing a numeric SLP-like program in which the value of any gate in the SLP is bounded in $[-1, 1]$. This will be used to prove our hardness results for bounded norms and show that the definition of SLPs given in Theorem 2.2 can indeed be set to a rational constant (rather than always 1) without the loss of generality.

**Definition A.3** (Bounded-Norm SLP (BN-SLP)). An instance is a numeric SLP $Q = (b_0, b_1, \ldots, b_m)$ of length $m$ over operations $\{+, -, \times\}$ whose *constant* is $b_0 := 2^{-m}$.

Equivalently, this SLP is evaluated at $x = 2^{-m}$ instead of $x = 1$. Thus, every gate value is a *rational* number. Moreover, $b_0 = 1/2^m$ has a standard binary encoding of $O(m)$ bits.

**Lemma A.4.** Let $Q = (b_0, b_1, \ldots, b_m)$ be a BN-SLP instance. Then, for every $i \in \{0, 1, \ldots, m\}$ we have $b_i \in [-1, 1]$.

*Proof.* Let $M_i := \max_{t \leq i} |b_t|$. We prove by induction that $M_i \leq 2^i b_0$ for all $i$, which implies $|b_i| \leq M_m \leq 2^m b_0 = 1$. The base case is $M_0 = |b_0| = b_0 \leq 2^0 b_0$. For the induction step, fix $i \geq 1$. If $b_i = b_j \pm b_k$ then

$|b_i| \leq |b_j| + |b_k| \leq 2M_{i-1} \leq 2^i b_0$. If $b_i = b_j \cdot b_k$ then $|b_i| \leq M_{i-1}^2 \leq (2^{i-1} b_0)^2 = 2^{2i-2} b_0^2 \leq 2^i b_0$, since $2^{i-2} b_0 \leq 2^{m-2} \cdot 2^{-m} = 2^{-2} \leq 1$. Thus $M_i \leq 2^i b_0$ for all $i$. $\square$

In the following, we show that an SLP can be effectively reduced to BN-SLP up to an exponential scale factor. This reduction cannot be done simply by scaling the first variable, since the addition/subtraction and multiplication combined would not scale correctly.

**Lemma A.5** (Reduction from SLP to BN-SLP). *There exists a deterministic polynomial-time reduction that, given an SLP $P$ of length $n$ computing an integer $n_P$ on $x = 1$, outputs a BN-SLP instance $I(P)$ of length $m = \mathsf{poly}(n)$ that computes the rational $2^{-m \cdot 2^n} \cdot n_P$.*

*Proof.* Let $P = (a_0, a_1, \ldots, a_n)$ be an SLP with $a_0 = 1$ (i.e., $a_1(1) = a_0 = 1$) where each gate $a_i$ for $i \geq 1$ is of the form $a_j \oplus a_k$ for some $0 \leq j, k < i$ and $\oplus \in \{+, -, \times\}$. We construct a BN-SLP instance by an SLP $Q$. We define the reduction in two passes. First, given $P$, we compute (in polynomial time) the exact number $m = m(P) = O(n^2)$ of gates that the construction below will output, by performing the same exponent bookkeeping and counting only the number of arithmetic gates added (ignoring the numeric value of $b_0$). Then we set $b_0 := 2^{-m}$ and emit exactly $m$ gates accordingly. Since $m(P)$ depends only on the syntactic structure of $P$ and the gate-counting procedure, this definition is non-circular and requires no padding.

We construct an SLP $Q$ using the constant $b_0$ and the operations $\{+, -, \times\}$. For the construction, we use integers $e_i \geq 1$ (not used directly in the SLP) for $i \in \{0, 1, \ldots, n\}$, such that for each original gate $a_i$ we compute a corresponding value $\tilde{a}_i$ satisfying the invariant

$$\tilde{a}_i = b_0^{e_i} a_i. \tag{1}$$

We set $\tilde{a}_0 := b_0$ and $e_0 := 1$, so (1) holds for $i = 0$ since $a_0 = 1$. We give an inductive construction of $\tilde{a}_i$ and $e_i$ for $i \geq 1$.

*Multiplication.* If $a_i = a_j \cdot a_k$, define $\tilde{a}_i := \tilde{a}_j \cdot \tilde{a}_k$ and $e_i := e_j + e_k$. Then (1) holds for $i$.

*Addition/Subtraction.* If $a_i = a_j \pm a_k$, let $E := \max(e_j, e_k)$ and define

$$\tilde{a}'_j := \tilde{a}_j \cdot b_0^{E-e_j}, \qquad \tilde{a}'_k := \tilde{a}_k \cdot b_0^{E-e_k},$$

(where $b_0^0 := 1$ is implemented by leaving the value unchanged), and set $\tilde{a}_i := \tilde{a}'_j \pm \tilde{a}'_k$ and $e_i := E$. Then (1) holds for addition and subtraction as well. The exponents satisfy $e_0 = 1$ and, for $i \geq 1$, either $e_i \leq \max(e_j, e_k)$ (addition/subtraction) or $e_i = e_j + e_k$ (multiplication). Therefore $e_i \leq 2^i$ for all $i$ by induction. We define the output node

$\tilde{a}_{n+1}$ of $Q$ receiving $b_0^{2^n - e_n} \cdot \tilde{a}_n$, giving $e_{n+1} = 2^n$ (if $e_n = 2^n$ we avoid adding this extra node).

*Implementing multiplication by $b_0^t$.* Fix an integer $t \geq 1$ and let $t = \sum_{i=0}^{\lfloor \log_2 t \rfloor} t_i 2^i$ with $t_i \in \{0, 1\}$. We construct an SLP gadget $G_t$ over the sole constant $b_0$ and operation $\times$ such that, given an input gate value $z$, it outputs $z \cdot b_0^t$ and has size $O(\log t)$. Specifically, the gadget first computes $p_0 := b_0$ and $p_{r+1} := p_r \cdot p_r = b_0^{2^{r+1}}$ for $r = 0, \ldots, \lfloor \log_2 t \rfloor - 1$. Then it sequentially multiplies $z$ by each $p_i$ with $t_i = 1$, yielding $z \cdot b_0^t$.

*Size bound.* Since $e_i \leq 2^i$ for all $i$, every alignment exponent $t \in \{E - e_j, E - e_k\}$ is at most $2^n$. Thus, each alignment multiplication by $b_0^t$ costs $O(\log t) = O(n)$ gates, and the total size is $m = O(n^2)$.

*Correctness.* By (1), the simulated output satisfies $\tilde{a}_n = b_0^{e_n} a_n$ and therefore $\tilde{a}_{n+1} = b_0^{2^n} \cdot n_P$ as required. Moreover, by Lemma A.4, every gate value in $I(P)$ lies in $[-1, 1]$. $\square$

By Lemma A.5, we can reduce BitSLP and PosSLP to rational bounded-norm variants, whose all intermediate values lie in $[-1, 1]$. The hardness of BitSLP and PosSLP given in Theorem 2.4 and Theorem 2.6, respectively, is preserved for the BN-SLP variants since multiplying an integer by a small positive rational $2^{-m2^n}$ preserves the sign and preserves all bit information shifted to the right by $m2^n$ bits. Namely, it remains hard to decide the $j$-th bit of the SLP output, where $j$ one of the first $n$ bits to the right of the decimal point. In addition, the positivity of an SLP output number is preserved by multiplying by a small factor; hence, Theorem 2.6 holds for BN-SLP instances as well. Therefore, we apply Theorem 2.4 and Theorem 2.6 for regular SLP and BN-SLP instances interchangeably.

## B. Deferred Proofs from Section 3

We prove below the main result of this section.

PROOF OF LEMMA 3.2

*Proof.* By Lemma 3.1 applied to $f = \sigma \in \mathbb{Q}[T]$, there is a polynomial time algorithm that, given $\sigma$, returns $\lambda_0, \ldots, \lambda_\mu \in \mathbb{Q}$ such that for all $x, y \in \mathbb{Q}$ it holds that

$$\sum_{j=0}^{\mu} \lambda_j \big( \sigma(x + y + j) - \sigma(x + j) - \sigma(y + j) \big) = xy. \quad (2)$$

Let $D$ be a common denominator of all the coefficients of $\sigma$ and all $\lambda_j$ (as rationals). Multiplying (2) by $D$ gives

$$Dxy = \sum_{j=0}^{\mu} \lambda'_j \big( \sigma(x + y + j) - \sigma(x + j) - \sigma(y + j) \big), \quad (3)$$

where $\lambda'_j := D\lambda_j \in \mathbb{Z}$.

**Construction of the network.** We now construct a directed acyclic network $N = (V, E)$, together with activations $\sigma^v$. We also construct a parameter vector $\theta^*$ that later will be shown to be the unique optimum.

*Vertices.*

- For each $i \in \{0, \ldots, \ell\}$ create a vertex $v_i$. The source is $s := v_0$ and the output is $t := v_\ell$.

- For each gate $i$ in the SLP with $a_i = a_j \cdot a_k$ (a multiplication gate), and for each $r \in \{0, \ldots, \mu\}$, create six auxiliary vertices:

$$L_{i,r}^{(1)}, \ L_{i,r}^{(2)}, \ L_{i,r}^{(3)}, \ U_{i,r}^{(1)}, \ U_{i,r}^{(2)}, \ U_{i,r}^{(3)},$$

and one additional vertex $z_i$.

*Activations.*

- For all $v \in \{v_0, \ldots, v_\ell\}$ and for all $L_{i,r}^{(t)}$ and $z_i$, set $\sigma^v(z) := z$ (identity).

- For all $U_{i,r}^{(t)}$, set $\sigma^v := \sigma$.

*Edges and the distinguished parameter vector $\theta^*$.* We define the edge set $E$ below. In addition, we define a *distinguished* parameter vector $\theta^* = ((w_{u,v}^*, b_{u,v}^*))_{(u,v) \in E} \in \Theta$ by specifying, for each edge $(u, v) \in E$, its target weight $w_{u,v}^* \in \mathbb{Q}$ and bias $b_{u,v}^* \in \mathbb{Q}$ as follows. Note that in the ERM instance, all edge-parameters are trainable; $\theta^*$ is used only as the target solution in the analysis and for defining auxiliary samples.

- *Source.* For $s = v_0$ we take $f^s(x) = x$ by definition (no incoming edges).

- *Addition / subtraction gates.* For each gate $i$ with $a_i = a_j + a_k$ and $j, k < i$:
  - Add edges $e_{j \to i} = (v_j, v_i)$ and $e_{k \to i} = (v_k, v_i)$.
  - Set $w_{e_{j \to i}}^* = 1$, $w_{e_{k \to i}}^* = 1$.
  - Set $b_{e_{j \to i}}^* = b_{e_{k \to i}}^* = 0$.

  For each gate $i$ with $a_i = a_j - a_k$:
  - Add edges $e_{j \to i} = (v_j, v_i)$ and $e_{k \to i} = (v_k, v_i)$.
  - Set $w_{e_{j \to i}}^* = 1$, $w_{e_{k \to i}}^* = -1$.
  - Set $b_{e_{j \to i}}^* = b_{e_{k \to i}}^* = 0$.

- *Multiplication gates.* For each gate $i$ with $a_i = a_j \cdot a_k$ and $j, k < i$, and for each $r \in \{0, \ldots, \mu\}$:
  - Add edges from $v_j$ and $v_k$ to $L_{i,r}^{(1)}$, both with weight $w_{v_j \to L_{i,r}^{(1)}}^* = 1$, $w_{v_k \to L_{i,r}^{(1)}}^* = 1$, and add a bias $r$ (encode this as a bias on one of the

incoming edges, chosen arbitrarily, such that $b^*_{v_j \to L^{(1)}_{i,r}} + b^*_{v_k \to L^{(1)}_{i,r}} = r$). Thus $f^{L^{(1)}_{i,r}}(a_0) = f^{v_j}(a_0) + f^{v_k}(a_0) + r$.

- Add an edge from $v_j$ to $L^{(2)}_{i,r}$ with weight $w^*_{v_j \to L^{(2)}_{i,r}} = 1$ and bias $b^*_{v_j \to L^{(2)}_{i,r}} = r$, so $f^{L^{(2)}_{i,r}}(a_0) = f^{v_j}(a_0) + r$.

- Add an edge from $v_k$ to $L^{(3)}_{i,r}$ with weight $w^*_{v_k \to L^{(3)}_{i,r}} = 1$ and bias $b^*_{v_k \to L^{(3)}_{i,r}} = r$, so $f^{L^{(3)}_{i,r}}(a_0) = f^{v_k}(a_0) + r$.

- Add edges $\left(L^{(t)}_{i,r}, U^{(t)}_{i,r}\right)$ with weight $w^*_{L^{(t)}_{i,r} \to U^{(t)}_{i,r}} = 1$ and bias $b^*_{L^{(t)}_{i,r} \to U^{(t)}_{i,r}} = 0$ for $t \in \{1,2,3\}$.

- For all $t \in \{1,2,3\}$ Add an edge from $U^{(t)}_{i,r}$ to $z_i$ with weight

$$
w^*_{U^{(t)}_{i,r} \to z_i} = \begin{cases} \lambda'_r & \text{if } t = 1, \\ -\lambda'_r & \text{if } t = 2,3. \end{cases}
$$

with zero bias $b^*_{U^{(t)}_{i,r} \to z_i} = 0$.

Finally, add an edge $(z_i, v_i)$ with weight $w^*_{z_i \to v_i} = D^{-1}$ and bias $b^*_{z_i \to v_i} = 0$ for each multiplication gate $i$ (i.e., $a_i = a_j \cdot a_k$).

Since $\sigma$ and the coefficients $\lambda_j$ are fixed, $D$ and all $\lambda'_r \in \mathbb{Z}$ are fixed rational constants with bounded bit-length, so the weights and biases in $\theta^*$ have bit-length $O_\sigma(1)$ depending only on $\sigma$ (and not on the SLP instance encoding size). Thus, the encoding size of the resulting instance $N$ and $\theta^*$ is polynomial in the length of $P$.

**Correctness of the SLP simulation under $\theta^*$.** In the ERM model below the input is a vector $x \in \mathbb{Q}^V$ that injects additively at every vertex. Whenever we write $f^v_{\theta^*}(a_0)$ in this simulation part, we mean $f^v_{\theta^*}(x)$ for the special input $x$ with $x_{v_0} = a_0$ and $x_v = 0$ for all $v \neq v_0$ (later defined in the ERM instance as $x_{\text{main}}$).

We prove by induction on $i$ that it holds that

$$
f^{v_i}(a_0) = a_i \quad \text{for all } i \in \{0, \dots, \ell\}.
$$

*Base case:* $i = 0$. We have $f^{v_0}(a_0) = f^s(a_0) = a_0$.

*Induction step:* Assume $f^{v_j}(a_0) = a_j$ for all $j < i$. Consider a gate $i$ with $a_i = a_j \oplus a_k$ and $j, k < i$.

If $\oplus = +$, then by construction and induction hypothesis,

$$
f^{v_i}(a_0) = f^{v_j}(a_0) + f^{v_k}(a_0) = a_j + a_k = a_i.
$$

If $\oplus = -$, the same argument with weights $1$ and $-1$ gives $f^{v_i}(a_0) = a_j - a_k = a_i$.

If $\oplus = *$, then by the induction hypothesis $f^{v_j}(a_0) = a_j$ and $f^{v_k}(a_0) = a_k$. Let $x := f^{v_j}(a_0)$ and $y := f^{v_k}(a_0)$, so $x = a_j$ and $y = a_k$. For each $r \in \{0, \dots, \mu\}$ we have

$$
f^{L^{(1)}_{i,r}}(a_0) = x + y + r, \ f^{L^{(2)}_{i,r}}(a_0) = x + r, \ f^{L^{(3)}_{i,r}}(a_0) = y + r.
$$

In addition by construction,

$$
f^{U^{(t)}_{i,r}}(a_0) = \sigma\big(f^{L^{(t)}_{i,r}}(a_0)\big) \quad \text{for } t \in \{1,2,3\}.
$$

Then, by the definition of $z_i$,

$$
f^{z_i}(a_0) = \sum_{r=0}^{\mu} \lambda'_r \Big(f^{U^{(1)}_{i,r}}(a_0) - f^{U^{(2)}_{i,r}}(a_0) - f^{U^{(3)}_{i,r}}(a_0)\Big)
$$
$$
= \sum_{r=0}^{\mu} \lambda'_r \Big(\sigma(x+y+r) - \sigma(x+r) - \sigma(y+r)\Big).
$$

Applying (3) yields $f^{z_i}(a_0) = D a_j a_k$. Finally, the edge $(z_i, v_i)$ has weight $D^{-1}$ and identity activation, so

$$
f^{v_i}(a_0) = D^{-1} f^{z_i}(a_0) = a_j \cdot a_k = a_i.
$$

This completes the induction. In particular,

$$
f_{\theta^*}(a_0) = f^t(a_0) = a_\ell = n_P. \tag{4}
$$

**Defining the ERM instance.** In the constructed ERM instance, *every vertex* is both an input coordinate and an output coordinate. Thus, the input space is $\mathcal{X} = \mathbb{Q}^V$ and the predictor outputs the full vector $f_\theta(x) \in \mathbb{Q}^V$, where for a topological order of $V$ the node values are computed, for every $v \in V$ by

$$
f^v_\theta(x) := \sigma^v\Big(x_v + \sum_{(u,v) \in E} \big(w_{u,v} f^u_\theta(x) + b_{u,v}\big)\Big). \tag{5}
$$

Fix a promise gap $a < b$ (with $a, b \in \mathbb{N}$). Let $j$ be the queried bit index from the BitSLP instance. We define the label space as $\mathcal{Y} := \mathbb{Q}^V \times \{0,1\}$, where the second component is a flag indicating whether the sample is auxiliary (i.e., 0) or main (i.e., 1).

We use the following efficiently computable bit extractor on rationals: write $q = u/v$ with $u \in \mathbb{Z}$, $v \in \mathbb{N}$ in lowest terms and define

$$
\text{bit}_j(q) := \left\lfloor 2^{-j} \frac{|u|}{v} \right\rfloor \bmod 2.
$$

Observe that this is computable in time polynomial in the bit-length of $(u, v, j)$.

Define the loss $\mathcal{L} : \mathbb{Q}^V \times (\mathbb{Q}^V \times \{0,1\}) \to \{0,1\}$ by[3]

$$
\mathcal{L}(\hat{y}, (y, \text{flag})) := \begin{cases} \mathbf{1}\big[\text{bit}_j(\hat{y}_t) \neq 1\big], & \text{if flag} = 1, \\ \mathbf{1}\big[\hat{y} \neq y\big], & \text{if flag} = 0. \end{cases}
$$

---

[3]Without the loss of generality, we avoid producing a flag for the output of the network; by adding an arbitrary flag to the output, we can define the loss as $\mathcal{L} : \mathcal{Y} \times \mathcal{Y} \to \{0,1\}$

In the above, recall that $t$ is the target node of the network. Note that equality of rational vectors is polynomial-time decidable by coordinatewise rational equality.

**Auxiliary samples.** We use auxiliary samples that force the trainable parameter vector $\theta$ to implement the distinguished vector $\theta^*$ (up to $\sigma$-symmetries on $\sigma$-edges). Recall that $\mu = \deg(\sigma) \geq 2$. Since $\sigma$ is a nonconstant polynomial, the polynomial $\sigma(z) - \sigma(0)$ has at most $\mu$ roots, so there exists $\alpha_1 \in \{1, 2, \ldots, \mu + 1\}$ with $\sigma(\alpha_1) \neq \sigma(0)$; fix such an $\alpha_1$. Set $\Delta_\sigma := \sigma(\alpha_1) - \sigma(0) \neq 0$.

Define the *baseline output template* $y^{(0)} \in \mathbb{Q}^V$ by

$$y_v^{(0)} := \begin{cases} 0, & \text{if } \sigma^v = \text{id}, \\ \sigma(0), & \text{if } \sigma^v = \sigma \end{cases}$$

for every $v \in V$. For auxiliary constructions we will specify, for each template, not only the desired node-values $y \in \mathbb{Q}^V$ but also a *preactivation template* $\tilde{y} \in \mathbb{Q}^V$ such that

$$y_v = \begin{cases} \tilde{y}_v, & \text{if } \sigma^v = \text{id}, \\ \sigma(\tilde{y}_v), & \text{if } \sigma^v = \sigma. \end{cases}$$

Given such a pair $(y, \tilde{y})$, define an input $x(y, \tilde{y}) \in \mathbb{Q}^V$ which for every $v \in V$ is

$$x(y, \tilde{y})_v := \tilde{y}_v - \sum_{(u,v) \in E} \left( w_{u,v}^* y_u + b_{u,v}^* \right). \quad (6)$$

By a direct induction along any topological order of $V$, the choice (6) enforces

$$f_{\theta^*}\left( x(y, \tilde{y}) \right) = y. \quad (7)$$

Moreover, in all auxiliary templates below, every coordinate of $y$ and $\tilde{y}$ is an explicit rational obtained from the constants

$$\{0, 1, \ldots, \mu\} \cup \{\sigma(0), \sigma(\alpha_1)\} \cup \sigma(\{0, 1, \ldots, \mu\})$$

together with coefficients appearing in $\theta^*$ (namely in $\{D^{-1}, \pm 1, \pm \lambda_r'\}$) using only $O_\sigma(1)$ many additions and multiplications (since every indegree is $O_\sigma(1)$). Hence each coordinate of $x(y, \tilde{y})$ defined in (6) has at most polynomial bit-length, and the total encoding size of each auxiliary example is polynomial in $|P|$.

Let $E_{\text{id}} := \{(u, v) \in E : \sigma^v = \text{id}\}$, let $\mathcal{M}$ be the set of all multiplication gates, and let

$$E_\sigma := \left\{ \left( L_{i,r}^{(t)}, U_{i,r}^{(t)} \right) : i \in \mathcal{M}, r \in \{0, 1 \ldots, \mu\}, t \in [3] \right\}$$

where $[3] = \{1, 2, 3, \}$. The latter set contains exactly the edges whose head is a $\sigma$-node.

Define the baseline preactivation template $\tilde{y}^{(0)} \in \mathbb{Q}^V$ by $\tilde{y}_v^{(0)} := 0$ for all $v \in V$ (so $y_v^{(0)} = \sigma^v \left( \tilde{y}_v^{(0)} \right)$ for every $v$).

For each edge $e = (p, q) \in E_{\text{id}}$, define a template pair $(y^{(e)}, \tilde{y}^{(e)})$ as follows:

$$y_v^{(e)} := y_v^{(0)} \qquad \text{for all } v \notin \{p, q\},$$

$$y_p^{(e)} := \begin{cases} 1, & \text{if } \sigma^p = \text{id}, \\ \sigma(\alpha_1), & \text{if } \sigma^p = \sigma, \end{cases}$$

$$y_q^{(e)} := w_{p,q}^* \left( y_p^{(e)} - y_p^{(0)} \right),$$

and set the preactivation template by

$$\tilde{y}_v^{(e)} := \begin{cases} y_v^{(e)}, & \text{if } \sigma^v = \text{id}, \\ 0, & \text{if } \sigma^v = \sigma \text{ and } v \neq p, \\ \alpha_1, & \text{if } \sigma^p = \sigma \text{ and } v = p. \end{cases}$$

Note that $y_p^{(e)} - y_p^{(0)}$ is either 1 or $\Delta_\sigma$, hence nonzero.

For each $\sigma$-edge $e_\sigma = (L, U) \in E_\sigma$ and each $\tau \in \{0, 1, \ldots, \mu\}$, define a template pair $(y^{(e_\sigma, \tau)}, \tilde{y}^{(e_\sigma, \tau)})$ by:

$$y_v^{(e_\sigma, \tau)} := y_v^{(0)} \quad \text{for all } v \notin \{L, U\},$$

$$y_L^{(e_\sigma, \tau)} := \tau,$$

$$y_U^{(e_\sigma, \tau)} := \sigma(\tau),$$

$$\tilde{y}_v^{(e_\sigma, \tau)} := 0 \quad \text{for all } v \notin \{L, U\},$$

$$\tilde{y}_L^{(e_\sigma, \tau)} := \tau,$$

$$\tilde{y}_U^{(e_\sigma, \tau)} := \tau.$$

(Then $x(y^{(e_\sigma, \tau)}, \tilde{y}^{(e_\sigma, \tau)})_U = 0$ by (6), since $U$ has the single incoming edge $(L, U)$ with $w_{L,U}^* = 1$ and $b_{L,U}^* = 0$.)

Define the auxiliary multiset

$$\mathcal{D}_a := \left\{ (x(y^{(0)}, \tilde{y}^{(0)}), (y^{(0)}, 0)) \right\}^{\times (b+1)}$$

$$\cup \bigcup_{e \in E_{\text{id}}} \left\{ (x(y^{(e)}, \tilde{y}^{(e)}), (y^{(e)}, 0)) \right\}^{\times (b+1)}$$

$$\cup \bigcup_{e_\sigma \in E_\sigma} \bigcup_{\tau=0}^{\mu} \left\{ (x(y^{(e_\sigma, \tau)}, \tilde{y}^{(e_\sigma, \tau)}), (y^{(e_\sigma, \tau)}, 0)) \right\}^{\times (b+1)}.$$

By (7), $\theta^*$ incurs zero loss on $\mathcal{D}_a$.

**Main samples.** Define the main input $x_{\text{main}} \in \mathbb{Q}^V$ by

$$x_{\text{main}}(v_0) = a_0, \qquad x_{\text{main}}(v) = 0 \ (v \neq v_0).$$

Let $y_{\text{main}} \in \mathbb{Q}^V$ be arbitrary (e.g. 0 everywhere), and set the main label to $(y_{\text{main}}, 1)$. Let $\mathcal{D}_m$ contain $b$ identical copies of $(x_{\text{main}}, (y_{\text{main}}, 1))$.

Finally, define the overall dataset as $\mathcal{D} := \mathcal{D}_a \cup \mathcal{D}_m$. We have the following claim used for proving the auxiliary samples force zero loss solutions to satisfy $\theta$ has similar attributes to $\theta^*$.

**Claim B.1.** *If $\theta$ satisfies*

$$\sum_{(x,(y,\mathrm{flag}))\in\mathcal{D}_a} \mathcal{L}(f_\theta(x),(y,\mathrm{flag})) = 0,$$

*then:*

- *For every identity-head edge $(p,q) \in E_{\mathrm{id}}$, one has $w_{p,q} = w_{p,q}^*$ and $\sum_{(u,q)\in E} b_{u,q} = \sum_{(u,q)\in E} b_{u,q}^*$.*

- *For every $\sigma$-edge $(L,U) \in E_\sigma$, the edge parameters satisfy the polynomial identity $\sigma(w_{L,U}z + b_{L,U}) \equiv \sigma(z)$ as polynomials in $z$; thus, on every input $x$ one has $f_\theta^U(x) = \sigma(f_\theta^L(x))$.*

*Proof.* Zero auxiliary loss implies that for every auxiliary template used above,

$$f_\theta(x(y,\tilde{y})) = y.$$

*Identity-head edges.* Fix $e = (p,q) \in E_{\mathrm{id}}$ and recall the auxiliary samples $(y^{(0)}, \tilde{y}^{(0)})$ and $(y^{(e)}, \tilde{y}^{(e)})$. Since $\sigma^q =$ id, by (5) and the equalities $f_\theta(x(y^{(0)}, \tilde{y}^{(0)})) = y^{(0)}$ and $f_\theta(x(y^{(e)}, \tilde{y}^{(e)})) = y^{(e)}$, we have

$$0 = y_q^{(0)} = x(y^{(0)}, \tilde{y}^{(0)})_q + \sum_{(u,q)\in E}\left(w_{u,q}\, y_u^{(0)} + b_{u,q}\right),$$

$$y_q^{(e)} = x(y^{(e)}, \tilde{y}^{(e)})_q + \sum_{(u,q)\in E}\left(w_{u,q}\, y_u^{(e)} + b_{u,q}\right).$$

By construction, $y_u^{(e)} = y_u^{(0)}$ for all $u \neq p,q$ and $x(y^{(e)}, \tilde{y}^{(e)})_q = x(y^{(0)}, \tilde{y}^{(0)})_q$ (this holds because $y_q^{(e)} = w_{p,q}^*(y_p^{(e)} - y_p^{(0)})$ was chosen exactly to keep the $q$-input unchanged). Subtracting the two displayed equations yields

$$y_q^{(e)} = w_{p,q}\left(y_p^{(e)} - y_p^{(0)}\right).$$

But by definition $y_q^{(e)} = w_{p,q}^*(y_p^{(e)} - y_p^{(0)})$ and $y_p^{(e)} - y_p^{(0)} \neq 0$ (it is either 1 or $\Delta_\sigma$), so $w_{p,q} = w_{p,q}^*$.

With $w_{u,q} = w_{u,q}^*$ for all $(u,q) \in E$ established by ranging over all $p$, plugging into the baseline equation for $q$ and using the definition of $x(y^{(0)}, \tilde{y}^{(0)})_q$ in (6) gives

$$\sum_{(u,q)\in E} b_{u,q} = \sum_{(u,q)\in E} b_{u,q}^*,$$

as claimed.

*$\sigma$-edges.* Fix $(L,U) \in E_\sigma$. For every $\tau \in \{0,1,\ldots,\mu\}$, the auxiliary sample corresponding to $(y^{(e_\sigma,\tau)}, \tilde{y}^{(e_\sigma,\tau)})$ enforces $f_\theta^L(x) = \tau$ and $f_\theta^U(x) = \sigma(\tau)$, and by construction $x(y^{(e_\sigma,\tau)}, \tilde{y}^{(e_\sigma,\tau)})_U = 0$ and $U$ has no incoming edges other than $(L,U)$. Thus, by (5) and the condition of the claim,

$$\sigma(w_{L,U}\tau + b_{L,U}) = \sigma(\tau) \qquad \text{for all } \tau \in \{0,1,\ldots,\mu\}.$$

Define $p(z) := \sigma(w_{L,U}z + b_{L,U}) - \sigma(z)$, a univariate polynomial of degree at most $\mu$. The above shows that $p$ has $\mu+1$ distinct roots, hence $p \equiv 0$, i.e., $\sigma(w_{L,U}z + b_{L,U}) \equiv \sigma(z)$ as polynomials. Consequently, for every input $x$,

$$f_\theta^U(x) = \sigma(w_{L,U}f_\theta^L(x) + b_{L,U}) = \sigma(f_\theta^L(x)).$$

This completes the proof of the claim. $\qquad\square$

Assume $\theta$ has zero auxiliary loss. By Claim B.1, all identity-head edge weights match $\theta^*$, and every $\sigma$-node $U_{i,r}^{(t)}$ satisfies $f_\theta^{U_{i,r}^{(t)}}(x) = \sigma(f_\theta^{L_{i,r}^{(t)}}(x))$ for all $x$. Therefore, the same induction as in the SLP simulation part applies to $x_{\mathrm{main}}$: addition/subtraction gates are computed with the correct weights w.r.t. $\theta^*$, and at each multiplication gate the gadget evaluates exactly the right expression in (3), so $f_\theta^t(x_{\mathrm{main}}) = n_P$.

**Completing the reduction.** Consider the empirical loss over $\mathcal{D}$. If $\theta$ has nonzero loss on any auxiliary sample, then since every auxiliary sample is repeated $(b+1)$ times,

$$\sum_{(x,(y,\mathrm{flag}))\in\mathcal{D}} \mathcal{L}(f_\theta(x),(y,\mathrm{flag})) \geq b+1 > b$$

so any $\theta$ with loss at most $b$ must have zero auxiliary loss. For such $\theta$, we have $f_\theta^t(x_{\mathrm{main}}) = n_P$ as above, and each main sample contributes loss 0 iff $\mathrm{bit}_j(n_P) = 1$, and loss 1 otherwise. Since there are $b$ identical main samples, the main-part loss is either 0 or $b$. Because $a < b$, there exists $\theta$ with total loss $\leq a$ iff $\mathrm{bit}_j(n_P) = 1$.

The construction is polynomial-time in $|P|$: $|V|, |E| = O(|P|)$, all weights/biases in $\theta^*$ have $O_\sigma(1)$ bit-length, and every auxiliary input $x(y,\tilde{y})$ is computed by (6) from constant-sized templates. This yields a polynomial-time reduction from BitSLP to $\mathrm{ERM}_{\mathrm{bit}}$. $\qquad\square$

*Remark* B.2 (Reducing both input and output dimensions to 1 via centered $L_2$ regularization). The above reduction can also be implemented with scalar input and scalar output ($\mathcal{X} = \mathcal{Y} = \mathbb{Q}$) by using a centered quadratic regularizer $\lambda\|\theta - \theta^*\|_2^2$ in the objective. The idea is to keep the same circuit network and use a single-bit loss on the output value (querying the $j$-th bit), replicated $b$ times.

At $\theta^*$, the network output is $n_P$, so the data term is 0 iff the $j$-th bit of $n_P$ is 1, and $b$ otherwise. Then choose $\lambda$ (as a dyadic rational $2^M$ with polynomial-size encoding) large enough so that every feasible $\theta \neq \theta^*$ contributes at least $b$ through $\lambda\|\theta - \theta^*\|_2^2$. Intuitively, this follows from the bit-model encoding bound: distinct feasible rational parameters are separated by an inverse-exponential gap, and $\lambda$ is picked to dominate that gap uniformly.

Hence any optimum is attained at $\theta^*$, and the YES/NO decision again matches whether the $j$-th bit of $n_P$ is 1. This

gives the same hardness with one-dimensional input and output.

As mentioned in the discussion, we explain in the following how to use a standard hinge loss instead of the bit-extraction loss to obtain a slightly weaker intractability result for $\text{ERM}_{\text{bit}}$

*Remark* B.3. Consider the (standard) hinge loss

$$\mathcal{L}_{\text{hinge}}(u, y) := \max\{0, 1 - yu\}, \qquad y \in \{+1, -1\}.$$

Assume we have the same training/architecture gadget as in the main reduction above, with a distinguished parameter vector $\theta^*$ and input $a_0$ such that the network output satisfies $f_{\theta^*}(a_0) = n_Q$ for an integer produced by an SLP $Q$, and with the same auxiliary forcing mechanism used in the main reduction to ensure that any optimizer equals $\theta^*$. We describe the following reduction from PosSLP to $\text{ERM}_{\text{bit}}$ with hinge loss. Given an SLP $P$ computing an integer $n_P$, construct an SLP $Q$ computing

$$n_Q := 2n_P - 1.$$

Then $n_Q \neq 0$ always (since $n_P \in \mathbb{Z}$), and

$$n_P > 0 \iff n_Q \geq 1, \text{ while } n_P \leq 0 \iff n_Q \leq -1.$$

Create an $\text{ERM}_{\text{bit}}$ instance with $b$ identical labeled samples $(x, y) = (a_0, +1)$, together with the same auxiliary forcing samples used in the main reduction so that the optimizer is pinned to $\theta^*$. Then the optimum value equals $\mathcal{L}_{\text{hinge}}(n_Q, +1) = \max\{0, 1 - n_Q\}$, hence:

$$n_Q \geq 1 \Rightarrow \text{OPT} = 0$$

and in addition

$$n_Q \leq -1 \Rightarrow \text{OPT} \geq 2.$$

Therefore, deciding whether $\text{OPT} \leq a$ or $\text{OPT} \geq b$ is PosSLP-hard with a more standard activation.

We now give the proof Lemma 3.1

PROOF OF THEOREM 3.1

*Proof.* Let $P_\mu := \mathbb{Q}[U]_{\leq \mu}$ be the $(\mu + 1)$-dimensional $\mathbb{Q}$-vector space of polynomials in one variable $U$ of degree at most $\mu$. For any $j = 0, 1, \ldots, \mu$, define $f_j(U) := f(U + j)$ and note that $f_j \in P_\mu$. We say that $B$ is a $\mathbb{Q}$-basis of $P_\mu$ if it is a basis of $P_\mu$ with coefficients taken only from $\mathbb{Q}$.

**Claim B.4.** *The family* $\{f_0, \ldots, f_\mu\}$ *is a* $\mathbb{Q}$*-basis of* $P_\mu$.

*Proof.* Since $\deg f_j = \mu$ for all $j$ and the dimension of the vector space $P_\mu$ over the rationals is $\dim_{\mathbb{Q}} P_\mu = \mu + 1$, it suffices to prove that $f_0, \ldots, f_\mu$ are linearly independent

over $\mathbb{Q}$. Suppose $\sum_{j=0}^{\mu} \alpha_j f_j(U) \equiv 0$ for some $\alpha_j \in \mathbb{Q}$ for all $j = 0, \ldots, \mu$. We show $\alpha_j = 0$ for all $j = 0, \ldots, \mu$. Recall that $f_j(U) = f(U + j) = \sum_{k=0}^{\mu} r_k(U + j)^k$, thus,

$$\begin{aligned}
0 &= \sum_{j=0}^{\mu} \alpha_j f_j(U) \\
&= \sum_{j=0}^{\mu} \alpha_j \sum_{k=0}^{\mu} r_k \cdot (U + j)^k \\
&= \sum_{k=0}^{\mu} r_k \sum_{j=0}^{\mu} \alpha_j (U + j)^k.
\end{aligned}$$

This is the zero polynomial in $U$; therefore, all its coefficients must be zero. We consider these coefficients from highest degree to lowest. We use this to prove the following claim by induction: We show below by induction that

$$\sum_{j=0}^{\mu} \alpha_j \cdot j^\ell = 0 \quad \text{for all } \ell = 0, 1, \ldots, \mu.$$

*Coefficient of $U^\mu$.* Only the $k = \mu$ term contributes to the coefficient of $U^\mu$ in the above. In addition, for each $j = 0, \ldots, \mu$ the coefficient of $U^\mu$ in $\sum_{k=0}^{\mu} r_k \sum_{j=0}^{\mu} \alpha_j (U + j)^k$ is $r_\mu \cdot \sum_{j=0}^{\mu} \alpha_j$. Since $r_\mu \neq 0$, we obtain $\sum_{j=0}^{\mu} \alpha_j = 0$. The above gives the base cases of the induction.

*Inductive step.* Assume that for some $m$ with $1 \leq m \leq \mu$ it holds that

$$\sum_{j=0}^{\mu} \alpha_j \cdot j^\ell = 0 \quad \text{for all } \ell = 0, 1, \ldots, m - 1.$$

Consider the coefficient of $U^{\mu-m}$ in $\sum_j \alpha_j f_j(U)$. By the binomial formula, for every $k \in \mathbb{N}$ it holds that

$$(U + j)^k = \sum_{q=0}^{k} \binom{k}{q} j^{k-q} U^q.$$

Thus, the coefficient of $U^{\mu-m}$ in $\sum_{j=0}^{\mu} \alpha_j f_j(U)$ is

$$\sum_{k=\mu-m}^{\mu} r_k \binom{k}{\mu - m} \sum_{j=0}^{\mu} \alpha_j \cdot j^{k-(\mu-m)}.$$

For $k < \mu$, we have $k - (\mu - m) \leq m - 1$, so by the inductive hypothesis $\sum_j \alpha_j \cdot j^{k-(\mu-m)} = 0$. Hence, all contributions with $k < \mu$ are zero, and only $k = \mu$ remains. Thus, the coefficient is:

$$r_\mu \cdot \binom{\mu}{\mu - m} \cdot \sum_{j=0}^{\mu} \alpha_j \cdot j^m = 0.$$

Since $r_\mu \neq 0$ and $\binom{\mu}{\mu-m} \neq 0$, we obtain

$$\sum_{j=0}^{\mu} \alpha_j \cdot j^m = 0.$$

By induction on $m = 0, 1, \ldots, \mu$, we conclude that

$$\sum_{j=0}^{\mu} \alpha_j j^\ell = 0 \quad \text{for all } \ell = 0, 1, \ldots, \mu.$$

Let $V$ be the Vandermonde matrix $V = (v_{\ell j})$ with $v_{\ell j} = j^\ell$ for $\ell, j = 0, \ldots, \mu$. The above conditions say exactly that $V\alpha = 0$, where $\alpha = (\alpha_0, \ldots, \alpha_\mu)^T$. By the known determinant of Vandermonde matrix formula, it holds that

$$\det V = \prod_{0 \leq i < j \leq \mu} (j - i).$$

Since $0, 1, \ldots, \mu$ are distinct, $\det V \neq 0$. Therefore, $V$ is invertible over $\mathbb{Q}$, implying that $\alpha = 0$. Therefore, $f_0, \ldots, f_\mu$ are linearly independent. Thus, $\{f_0, \ldots, f_\mu\}$ is a $\mathbb{Q}$-basis of $P_\mu$ as required. $\qquad\square$

Since $U^2 \in P_\mu$ and $\{f_0, \ldots, f_\mu\}$ is a basis of $P_\mu$, there exist unique $\lambda_0, \ldots, \lambda_\mu \in \mathbb{Q}$ such that

$$U^2 = \sum_{j=0}^{\mu} \lambda_j f(U + j). \tag{8}$$

The coefficients $\lambda_j$ indeed lie in $\mathbb{Q}$ because the change-of-basis matrix from $\{1, U, \ldots, U^\mu\}$ to $\{f_0, \ldots, f_\mu\}$ has entries in $\mathbb{Q}$ and is invertible over $\mathbb{Q}$, so its inverse also has entries in $\mathbb{Q}$.

Identity (8) holds in $\mathbb{Q}[U]$, so in $\mathbb{Q}[X, Y]$ we may substitute:

$$X^2 = \sum_{j=0}^{\mu} \lambda_j f(X + j), \tag{9}$$

$$Y^2 = \sum_{j=0}^{\mu} \lambda_j f(Y + j), \tag{10}$$

$$(X + Y)^2 = \sum_{j=0}^{\mu} \lambda_j f(X + Y + j). \tag{11}$$

In $\mathbb{Q}[X, Y]$ we have $X \cdot Y = \frac{(X+Y)^2 - X^2 - Y^2}{2}$. Using (9)–(11),

$$XY = \sum_{j=0}^{\mu} \frac{\lambda_j}{2} \big(f(X+Y+j) - f(X+j) - f(Y+j)\big). \tag{12}$$

Define

$$g(X, Y) = \sum_{j=0}^{\mu} \frac{\lambda_j}{2} \big(f(X+Y+j) - f(X+j) - f(Y+j)\big)$$

The above implies that for all $x, y \in \mathbb{R}$ it follows that $g(x, y) = x \cdot y$ with $\lambda_j' = \lambda_j/2$.

**Computing the coefficients $\lambda_j$ in polynomial time.** Assume $f(U) = \sum_{k=0}^{\mu} r_k U^k \in \mathbb{Q}[U]$ is given by its coefficients $r_k \in \mathbb{Q}$ in binary encoding. For each $j \in \{0, \ldots, \mu\}$ we can expand

$$f_j(U) = f(U + j) = \sum_{k=0}^{\mu} r_k(U + j)^k = \sum_{q=0}^{\mu} c_{q,j} U^q$$

using the binomial theorem, where each $c_{q,j} \in \mathbb{Q}$ is computable in time $\mathrm{poly}(\mu, \mathrm{bits}(f))$, denoting by $\mathrm{bits}(f)$ the number of bits needed to encode $f$. Let $C \in \mathbb{Q}^{(\mu+1)\times(\mu+1)}$ be the matrix whose $j$-th column is the coefficient vector $(c_{0,j}, \ldots, c_{\mu,j})^\top$ of $f_j$ in the basis $(1, U, \ldots, U^\mu)$. Then (8) is equivalent to the linear system

$$C\lambda = e_2,$$

where $\lambda = (\lambda_0, \ldots, \lambda_\mu)^\top$ and $e_2$ is the coefficient vector of $U^2$ (i.e., 1 in position 2 and 0 elsewhere). By Claim B.4, $C$ is invertible over $\mathbb{Q}$, hence the system has a unique solution in $\mathbb{Q}^{\mu+1}$. Solving a $(\mu+1) \times (\mu+1)$ linear system over $\mathbb{Q}$ (e.g. by Gaussian elimination with exact rational arithmetic) takes time $\mathrm{poly}(\mu, \mathrm{bits}(f))$, and produces $\lambda_0, \ldots, \lambda_\mu$ with polynomially bounded bit-length. $\qquad\square$

We can now prove our main result of this section.

*Proof of Theorem 1.1.* By Lemma 3.2, BitSLP Turing-reduces to $\mathrm{ERM}_{\mathrm{bit}}$. Thus, by Theorem 2.4 the proof follows. $\qquad\square$

## C. Deferred Proofs from Section 4

We now prove our hardness for Backprop-Sign and Backprop-Bit described in Theorem 1.3, holding even for an arbitrarily large promise gap (for PosSLP) or for all intermediate node values uniformly bounded by a constant (for BN-SLP). For brevity, we write one reduction for both of the above cases and distinguish between the cases when required.

PROOF OF THEOREM 1.3

*Proof.* We give polynomial-time reductions from PosSLP to Backprop-Sign and from BitSLP to Backprop-Bit. In the former, we define a Backprop-Sign instance with a promise gap $a < b$ for arbitrary fixed $a, b \in \mathbb{N}$. For the latter case, we simply define a Backprop-Bit instance. We give both constructions together for conciseness and highlight the differences when required.

Let $P = (a_0, \ldots, a_\ell)$ be an SLP. Applying Lemma 3.2 to $P$, we obtain a network $N_0 = (V_0, E_0)$, activations $\sigma^v \in \{\sigma, \mathrm{id}\}$, and rational parameters $\theta^0 = (w_e^0, b_e^0)_{e \in E_0}$ such that on input $x = a_0$,

$$f_{\theta^0}(a_0) = n_P := a_\ell \in \mathbb{Q}.$$

Let $t_0 \in V_0$ denote the output vertex of $N_0$. We form a new network $N = (V, E)$ as follows:

- Add a new target vertex $t$ with identity activation $\sigma^t(z) = z$.

- Add a new edge $e^\star := (t_0, t)$.

- Set $V := V_0 \cup \{t\}$ and $E := E_0 \cup \{e^\star\}$.

We fix all parameters on $E_0$ to their values in $\theta^0$, and set the bias on $e^\star$ to 0. The only free parameter will be the weight $w_{e^\star}$ on $e^\star$. For a parameter vector $\theta = \big((w_e, b_e)_{e \in E_0}, (w_{e^\star}, b_{e^\star})\big)$, with $(w_e, b_e) = (w_e^0, b_e^0)$ and $b_{e^\star} = 0$, the output on input $x = a_0$ is

$$f_\theta^t(a_0) = w_{e^\star} \cdot f_{\theta^0}^{t_0}(a_0) = w_{e^\star} \cdot n_P.$$

For defining the two reductions seamlessly, let $B$ be a variable defined as $B = 1$ in the reduction from BitSLP and $B = b$ in the reduction from PosSLP. We consider $B$ identical training samples

$$\mathcal{D} = \{(x_i, y_i)\}_{i \in [B]}, \qquad x_i = a_0, \quad y_i = -a_0.$$

We consider the standard square loss: $\mathcal{L} : \mathbb{Q} \times \mathbb{Q} \to \mathbb{Q}$ defined by
$$\mathcal{L}(x, y) := \tfrac{1}{2}(x - y)^2.$$

This loss is polynomial-time computable and differentiable in its first argument, and its partial derivative is $\partial \mathcal{L} / \partial x(x, y) = x - y$, which is polynomial-time computable and rational-valued on rational inputs. The total training loss is then

$$\begin{aligned} L(\theta) &= \sum_{(x,y) \in \mathcal{D}} \mathcal{L}\big(f_\theta(x), y\big) \\ &= \tfrac{B}{2}\big(f_\theta^t(a_0) - y\big)^2 \\ &= \tfrac{B}{2}\big(w_{e^\star} n_P + 1\big)^2. \end{aligned}$$

We view $L$ as a real-valued function of the real variables $(w_e, b_e)_{e \in E}$, defined by the same algebraic expression. Then for all such $\theta$ with the fixed sub network parameters and bias $b_{e^\star} = 0$, we have

$$\frac{\partial L}{\partial w_{e^\star}}(\theta) = B\big(f_\theta^t(a_0) - y\big) \frac{\partial f_\theta^t(a_0)}{\partial w_{e^\star}} = B\big(w_{e^\star} n_P + a_0\big) n_P.$$

In particular, for the specific parameter vector

$$\theta^* := \big((w_e^0, b_e^0)_{e \in E_0}, (w_{e^\star} = 0, b_{e^\star} = 0)\big),$$

since $w_{e^\star} = 0$ we have

$$\frac{\partial L}{\partial w_{e^\star}}(\theta^*) = B \cdot a_0 \cdot n_P.$$

We now consider two cases. If $P$ is a PosSLP instance, define the Backprop-Sign instance with promise corresponding to $P$ as

$$I(P) := (N, \mathcal{D}, \mathcal{L}, \theta^*, e^\star, b).$$

Otherwise, $P$ is a BitSLP instance and we define the Backprop-Bit instance corresponding to $P$ as

$$I(P) := (N, \mathcal{D}, \mathcal{L}, \theta^*, e^\star, j),$$

where $j \in \mathbb{N}$ is given in the input in binary. All rationals used in $N$ and $\theta^*$ (the parameters from $\theta^0$ and the fixed bias 0) have bit-length polynomial in the size of $P$ by Lemma 3.2; the loss $\mathcal{L}$ is given by a constant-size program, and $\mathcal{D} = \{(x_i, y_i)\}_{i \in [b]}$ also has polynomial bit-length encoding for fixed $b$. Hence, the total encoding size of $I(P)$ is polynomial in the encoding size of $P$, and the reduction $P \mapsto I(P)$ runs in polynomial time. By the above,

$$\frac{\partial L}{\partial w_{e^\star}}(\theta^*) = B \cdot a_0 \cdot n_P > 0 \iff n_P > 0.$$

Therefore, $P$ is a YES-instance of PosSLP if and only if $I(P)$ is a YES-instance of Backprop-Sign. This gives a polynomial-time reduction from PosSLP to Backprop-Sign. Note that if $P$ is a PosSLP instance, the gap $b \geq$ or $\leq -b$ is preserved if $a_0 = 1$.

In addition, considering a BitSLP instance $(j, P)$, then $B = 1$ and we can directly decide the $j$-th bit in $n_P$ by directly accessing it; if $a_0 = 1$, this is immediate; otherwise, we consider a BN-SLP with $a_0 = 2^{-\ell}$, which shifts the $j$-th bit of $n_P$ to the right by exactly $\ell$ bits. Therefore, we can recover the $j$-th bit of $n_P$ in polynomial time.

By the above and Theorem 2.4 it follows that Backprop-Bit is $\#P$-hard. In addition, if Backprop-Sign were in BPP, then PosSLP would be in BPP. By Theorem 2.6, under the Constructive Univariate Radical Conjecture (Dutta et al., 2022) together with PosSLP $\in$ BPP, it follows that NP $\subseteq$ BPP, contradicting our assumption. Thus, Backprop-Sign is not in BPP. $\qquad \square$

## D. PAC Learning in Real vs. Bit Models

In this section, we prove the following theorem.

**Theorem D.1.** *For every integer $q \geq 1$ there is a hypothesis class and distribution for which the realizable PAC sample complexity is $O(1)$ in the exact real-RAM model, but for every rounding precision $q$ in the bit model it is $\Omega(q)$.*

We begin with the formal definition of (realizable) PAC learning.

**Definition D.2** (Realizable PAC learning). Let $\mathcal{X}$ be a domain, $\mathcal{Y}$ a label space, and $\mathcal{H} \subseteq \mathcal{Y}^{\mathcal{X}}$ a hypothesis class.

Let $D$ be a distribution on $\mathcal{X}$. For $h \in \mathcal{H}$ and a hypothesis $\hat{h} : \mathcal{X} \to \mathcal{Y}$, the (0-1) error of $\hat{h}$ with respect to $h$ and $D$ is

$$\mathrm{err}_D(\hat{h}, h) := \Pr_{x \sim D}\left[\hat{h}(x) \neq h(x)\right].$$

A (possibly randomized) learning algorithm $\mathcal{A}$ is said to *realizably PAC learn* $(\mathcal{H}, D)$ with parameters $(\varepsilon, \delta)$ if there exists a sample size $m$ such that, for every target $h^\star \in \mathcal{H}$, whenever $\mathcal{A}$ is given $m$ i.i.d. labeled examples $(x_i, h^\star(x_i))$ with $x_i \sim D$, it outputs a hypothesis $\hat{h}$ satisfying

$$\Pr\left[\mathrm{err}_D(\hat{h}, h^\star) \leq \varepsilon\right] \geq 1 - \delta,$$

where the probability is over the draw of the sample and the internal randomness of $\mathcal{A}$.

We now define a simple one-parameter family of rounded multipliers at precision $2^{-q}$ and show that, even in the realizable setting under a fixed distribution, the number of samples needed to drive the error below $1/(q+1)$ grows linearly with $q$.

**Definition D.3** (Rounded multipliers at precision $2^{-q}$). Fix an integer $q \geq 1$ and define the domain

$$\mathcal{X}_q := \{2^0, 2^1, \ldots, 2^q\} \subset \mathbb{R}.$$

For $z \in \mathbb{R}$ let

$$\mathrm{round}_q(z) := 2^{-q}\left\lfloor z\, 2^q \right\rfloor,$$

i.e., rounding $z$ down to the nearest multiple of $2^{-q}$. For each real parameter $c \in [0, 1)$ define

$$h_c(x) := \mathrm{round}_q(cx), \quad x \in \mathcal{X}_q.$$

Let $\mathcal{H}_q := \{h_c : c \in [0, 1)\}$ and let $D_q$ be the uniform distribution on $\mathcal{X}_q$. We use 0-1 loss on the discrete labels $\mathcal{Y} := \{2^{-q}k : k \in \mathbb{Z}\}$, so that $\mathrm{err}_{D_q}(\hat{h}, h)$ is just the fraction of points in $\mathcal{X}_q$ where $\hat{h}$ and $h$ differ.

In the *exact real* model (no rounding) and noiseless setting, a single labeled example $(x, y)$ with $x \neq 0$ and $y = cx$ determines $c = y/x$ and thus $h_c$ on all of $\mathcal{X}_q$. The next theorem shows that under rounding at precision $2^{-q}$, even in the realizable PAC model under the fixed distribution $D_q$, the sample complexity to achieve error $< 1/(q+1)$ grows linearly with $q$.

**Theorem D.4.** *Fix $q \geq 1$ and consider the class $\mathcal{H}_q$ and distribution $D_q$ defined above. Let $\varepsilon < 1/(q+1)$ and let $\delta \in (0, 1/4)$. Then any (possibly randomized) algorithm that realizably* PAC *learns $(\mathcal{H}_q, D_q)$ with parameters $(\varepsilon, \delta)$ must use sample size $m \geq q \log \frac{1}{2\delta}$*

*Proof.* Fix $q \geq 1$. We first prove the lower bound for deterministic algorithms, and then extend to randomized algorithms by a standard argument.

We will exhibit two functions $h^{(1)}, h^{(2)} \in \mathcal{H}_q$ such that:

- $h^{(1)}(x) = h^{(2)}(x)$ for all $x \in \{2^0, \ldots, 2^{q-1}\}$,

- $h^{(1)}(2^q) \neq h^{(2)}(2^q)$.

As a consequence, any sample that does not contain the point $2^q$ looks identical under $h^{(1)}$ and $h^{(2)}$.

Write any $c \in [0, 1)$ in binary as $c = 0.b_1 b_2 b_3 \ldots$ with $b_i \in \{0, 1\}$ choosing one expansion arbitrarily when two exist. For $k \geq 0$,

$$c2^k = \sum_{i=1}^{k} b_i 2^{k-i} + \theta_k,$$

with $\theta_k \in [0, 1)$, hence $\lfloor c2^k \rfloor = \sum_{i=1}^{k} b_i 2^{k-i}$. Applying this with $k + q$ in place of $k$,

$$h_c(2^k) = \mathrm{round}_q(c2^k) = 2^{-q}\lfloor c2^{k+q} \rfloor = 2^{-q} \sum_{i=1}^{k+q} b_i 2^{k+q-i}.$$

Thus for $k \in \{0, \ldots, q-1\}$, $h_c(2^k)$ depends only on the first $k + q \leq 2q - 1$ bits of $c$.

Now fix any binary sequence $b_1, \ldots, b_{2q-1}$. Define $c^{(1)}, c^{(2)} \in [0, 1)$ by

$$c^{(1)} = 0.b_1 b_2 \ldots b_{2q-1} 000 \ldots,$$

and

$$c^{(2)} = 0.b_1 b_2 \ldots b_{2q-1} 100 \ldots,$$

and let $h^{(j)} := h_{c^{(j)}}$ for $j \in \{1, 2\}$. For any $k \in \{0, \ldots, q-1\}$ we have $k + q \leq 2q - 1$, so the first $k + q$ bits of $c^{(1)}$ and $c^{(2)}$ coincide, hence

$$h^{(1)}(2^k) = h^{(2)}(2^k) \quad \text{for all } k < q.$$

On the other hand, at $x^\star := 2^q$,

$$h^{(j)}(2^q) = 2^{-q}\lfloor c^{(j)} 2^{2q} \rfloor.$$

Since $c^{(1)}$ and $c^{(2)}$ agree on their first $2q - 1$ bits but differ at bit $2q$, we have

$$\lfloor c^{(2)} 2^{2q} \rfloor - \lfloor c^{(1)} 2^{2q} \rfloor = 1,$$

so $h^{(1)}(2^q) \neq h^{(2)}(2^q)$.

Fix a deterministic learning algorithm $\mathcal{A}$ and a sample size $m$. We will show that if $m < q \cdot \log(1/(2\delta))$, then there exists a target $h^\star \in \mathcal{H}_q$ (either $h^{(1)}$ or $h^{(2)}$) such that $\mathcal{A}$ cannot achieve error at most $\varepsilon < 1/(q+1)$ with probability at least $1 - \delta$.

Consider the following experiment: pick $h^\star$ uniformly at random from $\{h^{(1)}, h^{(2)}\}$ and draw an i.i.d. sample $S = ((x_i, y_i))_{i=1}^{m}$ with $x_i \sim D_q$ and $y_i = h^\star(x_i)$. Let $\hat{h}_S$ be the output of $\mathcal{A}$ on sample $S$.

Let $E$ be the event that the sample $S$ contains no occurrence of $x = 2^q$. Since $D_q$ is uniform on $\{2^0, \dots, 2^q\}$,

$$\Pr[E] = \left(1 - \tfrac{1}{q+1}\right)^m = \left(\tfrac{q}{q+1}\right)^m.$$

On the event $E$, all sampled points lie in $\{2^0, \dots, 2^{q-1}\}$, and we have seen that $h^{(1)}$ and $h^{(2)}$ coincide on these inputs. Hence, conditioned on $E$, the labeled sample $S$ has exactly the same distribution regardless of whether $h^\star = h^{(1)}$ or $h^\star = h^{(2)}$. Therefore, on $E$, the output $\hat{h}_S$ of the deterministic algorithm $\mathcal{A}$ is the same in both cases.

However, we have $h^{(1)}(2^q) \neq h^{(2)}(2^q)$, so on event $E$, that is, for every sample $S$ that does not contain $2^q$, *at least one* of the two possible targets leads to an error at $2^q$:

$$\mathbf{1}\{\hat{h}_S(2^q) \neq h^{(1)}(2^q)\} \ + \ \mathbf{1}\{\hat{h}_S(2^q) \neq h^{(2)}(2^q)\} \ \geq \ 1.$$

Taking expectation over the random choice of $h^\star$ (uniform on $\{h^{(1)}, h^{(2)}\}$) and of $S$, we obtain

$$\Pr_{h^\star, S}\!\left[\hat{h}_S(2^q) \neq h^\star(2^q)\right] \geq \tfrac{1}{2}\Pr[E]$$
$$= \tfrac{1}{2}\left(\tfrac{q}{q+1}\right)^m.$$

Since $D_q$ is uniform, note that for any fixed $h^\star \in \mathcal{H}_q$ and hypothesis $\hat{h}$,

$$\mathrm{err}_{D_q}(\hat{h}, h^\star) = \frac{1}{q+1} \sum_{x \in \mathcal{X}_q} \mathbf{1}\{\hat{h}(x) \neq h^\star(x)\},$$

so $\mathrm{err}_{D_q}(\hat{h}, h^\star) \in \{0, 1/(q+1), \dots, 1\}$. In particular, if $\hat{h}_S(2^q) \neq h^\star(2^q)$ then $\mathrm{err}_{D_q}(\hat{h}_S, h^\star) \geq 1/(q+1) > \varepsilon$, and hence

$$\Pr_{h^\star, S}\!\left[\mathrm{err}_{D_q}(\hat{h}_S, h^\star) > \varepsilon\right] \ \geq \ \tfrac{1}{2}\left(\tfrac{q}{q+1}\right)^m.$$

By averaging over the random choice of $h^\star \in \{h^{(1)}, h^{(2)}\}$, there exists a fixed $h^\star \in \mathcal{H}_q$ such that

$$\Pr_S\!\left[\mathrm{err}_{D_q}(\hat{h}_S, h^\star) > \varepsilon\right] \ \geq \ \tfrac{1}{2}\left(\tfrac{q}{q+1}\right)^m.$$

Requiring this probability to be at most $\delta$ and using $\log(1 + 1/q) \leq 1/q$ as $q > 0$ yields

$$\tfrac{1}{2}\left(\tfrac{q}{q+1}\right)^m \leq \delta \ \Rightarrow \ m \ \geq \ q \log\!\left(\tfrac{1}{2\delta}\right).$$

Any randomized learner is a distribution over deterministic learners (obtained by fixing its internal randomness). If a randomized learner with $m$ samples satisfied the $(\varepsilon, \delta)$ guarantee for all $h^\star \in \mathcal{H}_q$, then by averaging some deterministic learner in its support would also satisfy it, contradicting the lower bound above. Thus, the same lower bound on $m$ holds for randomized learners as well. □

# E. Efficient Backpropagation and ERM for Piecewise-Linear Activations

In this section, we prove Theorem 1.4, which follows from the two results stated in below combined with standard known NP-Hardness results for ReLU networks training.

In contrast to our hardness results for polynomial activations, standard piecewise-linear nonlinearities such as ReLU and leaky-ReLU do not introduce new multiplicative interactions in the bit model. In this section we show that, for such activations, both a single backpropagation step and verification of ERM objectives can be carried out in polynomial time, placing these problems in P (and hence $\mathsf{ERM}_{\mathrm{bit}}$ in NP).

**Bit-length conventions.** Let $N$ denote the total bit-length of the input encoding (architecture, parameters $\theta$, dataset $(x_i, y_i)_{i=1}^n$, and learning rate $\eta$). In particular, $n \leq N$, $|E| \leq N$, and every scalar appearing in the input (weights, biases, inputs, labels, $\eta$) has bit-length at most $N$. Recall that all arithmetic is over rationals in binary; if $a, b$ have bit-lengths at most $B, C$, then

$$\mathrm{len}(a \pm b) \leq \max\{B, C\} + 1, \qquad \mathrm{len}(ab) \leq B + C + 1.$$

**Theorem E.1** (One backpropagation step with piecewise–linear activations is in P)**.** *Assume all hidden and output neurons use piecewise-linear activations $\sigma^v : \mathbb{Q} \to \mathbb{Q}$ with rational breakpoints and rational slopes and intercept, and that each $\sigma^v$ and its derivative with respect to the input are computable in time polynomial in the bit-length of their arguments. Assume also that $\mathcal{L}$ and its derivative with respect to the network output are computable in time polynomial in the bit-length of their arguments.*

*Then one full gradient descent step for*

$$J(\theta) = \sum_{i=1}^n \mathcal{L}\big(f_\theta(x_i), y_i\big)$$

*computed by standard backpropagation runs in polynomial time in $|I|$.*

*Proof.* Note that the argument below uses only that each activation $\sigma^v$ (and its backprop rule/derivative) is computable in polynomial time on rational inputs and maps inputs of bit-length poly($N$) to outputs of bit-length poly($N$); therefore the same proof applies to any *bit-bounded* (e.g., $k$-bit rounded/clipped) activation family with $k \leq$ poly($N$).

*Operation count.* Standard backpropagation (forward pass, backward pass, gradient accumulation, and parameter update) uses $O(n \cdot |E|)$ additions, multiplications, and comparisons; since $n, |E| \leq N$, this is polynomially bounded in $N$.

*Bit-length growth for activations and errors.* Fix an example $(x_i, y_i)$. All preactivations $z_v^{(i)}$, activations $h_v^{(i)}$, and error terms $\delta_v^{(i)}$ are obtained from the input scalars (coordinates of $x_i, y_i$ and parameters $\theta$) by repeated application of:

- addition / subtraction of previously computed values,

- multiplication of a previously computed value by a *parameter* (weight or bias) or by a fixed rational slope or coefficient from some piece of a $\sigma^v$,

- evaluation of a piecewise-linear $\sigma^v$ or its derivative $\partial \sigma^v / \partial z$, which consists of a constant number of comparisons with rational breakpoints and then an affine expression of the form $az + b$ with $a, b \in \mathbb{Q}$.

All slopes, intercepts, and breakpoints of all $\sigma^v$ are part of the input and therefore have bit-length at most $N$. Thus, starting from values of bit-length at most $N$, along the computation of any $z_v^{(i)}$, $h_v^{(i)}$, or $\delta_v^{(i)}$ we perform at most $O(|E|)$ such arithmetic steps, each of which increases bit-length by at most $O(N)$ (multiplication by a rational of bit-length $\leq N$) or 1 (addition), while comparisons and case distinctions do not change bit-length. Hence every $z_v^{(i)}$, $h_v^{(i)}$, and $\delta_v^{(i)}$ has bit-length bounded by

$$N + O(|E|) \cdot (N + 1) \ \leq \ \mathrm{poly}(N).$$

*Bit-length of gradients and update.* For each edge $u \to v$,

$$\frac{\partial J}{\partial w_{u \to v}} = \sum_{i=1}^{n} \delta_v^{(i)} h_u^{(i)}, \qquad \frac{\partial J}{\partial b_v} = \sum_{i=1}^{n} \delta_v^{(i)}.$$

Each product $\delta_v^{(i)} h_u^{(i)}$ involves factors of bit-length $\mathrm{poly}(N)$ and therefore has bit-length $\mathrm{poly}(N)$; summing over $i = 1, \ldots, n$ adds at most $\log n \leq \log N$ bits. Thus every gradient entry has bit-length $\mathrm{poly}(N)$.

The parameter update $\theta \leftarrow \theta - \eta \nabla J(\theta)$ consists of multiplications of gradient entries by the input scalar $\eta$ (bit-length $\leq N$) and additions/subtractions with the previous parameters (bit-length $\leq N$), so the updated parameters also have bit-length bounded by $\mathrm{poly}(N)$.

*Conclusion.* During one backpropagation step, all intermediate values have bit-length $\mathrm{poly}(N)$. Each arithmetic operation on such numbers can be carried out in time polynomial in $N$ in our bit model. Since the number of operations is $O(n \cdot |E|) \leq O(N^2)$, the total running time is polynomial in $N$. Hence one backpropagation step with piecewise-linear activations lies in P. $\square$

We can now also prove Remark 1.5

**Proof of Remark 1.5** Identical to the proof above: the operation count is unchanged, and condition (ii) directly enforces that all intermediate quantities have bit-length $\mathrm{poly}(N)$, so each arithmetic operation costs $\mathrm{poly}(N)$ time.

**Theorem E.2.** *Assuming piecewise-linear activations, the* restricted ERM$_{\mathrm{bit}}$ *problem in the bit model is in* P. *In particular,* ERM *in the bit model (with explicit rational witnesses) lies in* NP.

*Proof.* In a restricted ERM$_{\mathrm{bit}}$ instance we are given a fixed parameter vector $\theta^*$ as part of the input. To decide the instance we must compute

$$\sum_{i=1}^{n} \mathcal{L}\big(f_{\theta^*}(x_i), y_i\big)$$

and compare it to $\gamma$.

This is exactly the forward part of the backpropagation computation (without the backward and update steps). As in the bit-length analysis above, each activation $h_v^{(i)}$ is obtained from inputs and parameters by a sequence of additions, multiplications by parameters or fixed slopes, and piecewise-linear activations of length at most $O(|E|)$, so all intermediate values (and hence each $f_{\theta^*}(x_i)$ and the sum of losses) have bit-length bounded by $\mathrm{poly}(N)$. The number of operations is $O(n \cdot |E|) \leq O(N^2)$, so this computation runs in polynomial time in $N$. Therefore, restricted ERM$_{\mathrm{bit}}$ with piecewise-linear activations in the bit model is in P.

Since, in the general ERM (bit-model) problem, a candidate parameter vector $\theta$ is given explicitly as a rational witness, and verifying its loss is in P by the same argument, ERM in the bit model lies in NP. $\square$

We note that NP-Completeness follows from the (numerous) known NP-Hardness results given in the literature (e.g., (Judd, 1988; Blum & Rivest, 1988; Šíma, 2002)). Thus, Theorem 1.4 follows from Theorem E.1, Theorem E.2, and the above NP-Hardness results.

## F. Not Every Rational Polynomial Can Represent $x \cdot y$

In this section, we study when a fixed univariate function $f$ can be used as a "universal gadget" to represent the product of two variables using only shifted evaluations of $f$ and rational coefficients. Concretely, we ask for identities of the form

$$xy \ = \ \sum_j \lambda_j \big( f(x + y + j) - f(x + j) - f(y + j) \big),$$

valid for all reals $x, y$, where the $\lambda_j$ are rational and only finitely many are nonzero.

We have shown in Theorem 3.1 that every nonlinear polynomial with rational coefficients admits such a representation. One might hope to extend this positive result from integer powers to non-integer rational exponents or negative integer exponents. The next theorem shows that this is impossible: for $f(T) = T^\alpha$ with $\alpha \in \mathbb{Q} \setminus \mathbb{Z}_{\geq 0}$, no finite rational combination of shifted copies of $f$ can represent $x \cdot y$.

**Theorem F.1.** *Let $\alpha \in \mathbb{Q} \setminus \mathbb{Z}_{\geq 0}$ and consider $f(T) = T^\alpha$ on $(0, \infty)$, with the principal real branch of $t^\alpha$. Then there do not exist an integer $m \geq 0$, rational coefficients $\lambda_0, \ldots, \lambda_m \in \mathbb{Q}$, and a real number $R > 0$ such that for all rational $x, y > R$,*

$$\sum_{j=0}^{m} \lambda_j \big( f(x+y+j) - f(x+j) - f(y+j) \big) = x \cdot y. \quad (13)$$

In particular, the representation of Lemma 3.1 for polynomials cannot be extended to $f(T) = T^\alpha$ with rational $\alpha \notin \mathbb{Z}_{\geq 0}$. To prove this result, we use the following.

**Lemma F.2.** *Let $\beta \in \mathbb{Q} \setminus \mathbb{Z}_{\geq 0}$ and let $b_0, \ldots, b_m \in \mathbb{Q}$ be pairwise distinct. Define*

$$\varphi_j(x) := (x + b_j)^\beta$$

*on an interval $(R, \infty)$ with $R > -\min_j b_j$, so that $x + b_j > 0$ for all $x > R$. If*

$$\sum_{j=0}^{m} \lambda_j \varphi_j(x) = 0 \quad \text{for all } x > R,$$

*for some $\lambda_0, \ldots, \lambda_m \in \mathbb{Q}$, then $\lambda_0 = \cdots = \lambda_m = 0$.*

*Proof.* Assume

$$\sum_{j=0}^{m} \lambda_j (x + b_j)^\beta = \sum_{j=0}^{m} \lambda_j \varphi_j(x) = 0 \quad \text{for all } x > R.$$

Set $t = 1/x$ for $x > R$; then for all sufficiently small $t > 0$, it holds that $x + b_j = 1/t + b_j > 0$; thus,

$$0 = \sum_{j=0}^{m} \lambda_j (1/t + b_j)^\beta.$$

Multiplying by $t^\beta$ gives

$$0 = t^\beta \sum_{j=0}^{m} \lambda_j (1/t + b_j)^\beta = \sum_{j=0}^{m} \lambda_j (1 + b_j t)^\beta.$$

Define

$$F(t) := \sum_{j=0}^{m} \lambda_j (1 + b_j t)^\beta.$$

For $|t|$ small, each $(1 + b_j t)^\beta$ admits a convergent binomial series, hence $F$ is analytic near $t = 0$. We have $F(t) = 0$

for all sufficiently small $t > 0$, so by analyticity $F(t) \equiv 0$ near 0.

Using the binomial expansion,

$$(1 + b_j t)^\beta = \sum_{k=0}^{\infty} \binom{\beta}{k} b_j^k t^k,$$

we obtain

$$0 = F(t) = \sum_{j=0}^{m} \lambda_j \sum_{k=0}^{\infty} \binom{\beta}{k} b_j^k t^k$$

$$= \sum_{k=0}^{\infty} \binom{\beta}{k} \Big( \sum_{j=0}^{m} \lambda_j b_j^k \Big) t^k.$$

Thus, since $F$ is analytic near 0 and $F(t) \equiv 0$, for every $k \geq 0$:

$$\binom{\beta}{k} \sum_{j=0}^{m} \lambda_j b_j^k = 0.$$

Since $\beta \notin \mathbb{Z}_{\geq 0}$, the binomial coefficient

$$\binom{\beta}{k} = \frac{\beta(\beta - 1) \cdots (\beta - k + 1)}{k!}$$

is nonzero for all $k \geq 0$, so

$$\sum_{j=0}^{m} \lambda_j b_j^k = 0 \quad \text{for all } k \geq 0.$$

For $k = 0, 1, \ldots, m$, this yields the homogeneous linear system $V\lambda = 0$, where $V$ is the $(m+1) \times (m+1)$ Vandermonde matrix $V_{kj} = b_j^k$. Since the $b_j$ are pairwise distinct, $\det V \neq 0$, so $V$ is invertible and hence $\lambda_0 = \cdots = \lambda_m = 0$. $\square$

We now give the proof of Theorem F.1

*Proof.* Assume towards a contradiction that such $m$, $\lambda_0, \ldots, \lambda_m$, and $R$ exist. For $x, y > R$ all arguments of $f$ in (13) are positive, so $f$ is real-analytic there. Both sides of (13) are analytic in $(x, y)$ on $(R, \infty)^2$ and agree on all rational pairs $(x, y)$ in this region, hence by analyticity the identity holds for all real $x, y > R$.

Differentiating (13) with respect to $x$ gives

$$y = \sum_{j=0}^{m} \lambda_j \alpha \Big( (x+y+j)^{\alpha-1} - (x+j)^{\alpha-1} \Big) \quad (14)$$

for all $x, y > R$. Differentiating (14) with respect to $y$ yields

$$1 = \sum_{j=0}^{m} \lambda_j \alpha(\alpha - 1)(x+y+j)^{\alpha-2} \quad (15)$$

for all $x, y > R$.

Let $z = x + y$. Then for all $z > 2R$, (15) becomes

$$1 = \alpha(\alpha - 1) \sum_{j=0}^{m} \lambda_j (z + j)^{\alpha - 2}. \qquad (16)$$

Differentiating (16) with respect to $z$ gives

$$0 = \alpha(\alpha - 1)(\alpha - 2) \sum_{j=0}^{m} \lambda_j (z + j)^{\alpha - 3} \qquad (17)$$

for all $z > 2R$.

Since $\alpha \in \mathbb{Q} \setminus \mathbb{Z}_{\geq 0}$, we have $\alpha, \alpha - 1, \alpha - 2 \neq 0$, so $\alpha(\alpha - 1)(\alpha - 2) \neq 0$. Let $\beta := \alpha - 3 \in \mathbb{Q} \setminus \mathbb{Z}_{\geq 0}$ and define $\lambda'_j := \alpha(\alpha - 1)(\alpha - 2)\lambda_j \in \mathbb{Q}$. Then

$$\sum_{j=0}^{m} \lambda'_j (z + j)^{\beta} = 0 \quad \text{for all } z > 2R.$$

By Lemma F.2 it follows that $\lambda_j = 0$ for all $j$. But then (13) reduces to $xy = 0$ for all $x, y > R$, a contradiction. Therefore no such finite representation (13) can exist. $\quad \square$

