# OpenReview forum: "Why ReLU? A Bit-Model Dichotomy for Deep Network Training"
_ICML.cc/2026/Conference — ICML 2026 regular_

### Official Review · Reviewer_zakj · 2026-02-21

**Soundness:** 2
**Presentation:** 2
**Significance:** 2
**Originality:** 2
**Overall Recommendation:** 3
**Confidence:** 1

**Summary:**

This paper conducted a hardness analysis of a neural network, which was previously covered in real parameter setting, in a rational number space with a finite number of bits. It demonstrates that problems with ReLU and polynomial activations have different levels of hardness.

**Compliance With Llm Reviewing Policy:**

Affirmed.

**Final Justification:**

The authors' response addressed my concerns. Still, I lack the expertise to decide the acceptance of this paper. Therefore, I raise my score to weak reject, but the AC may disregard this score. I will not object to this paper being accepted or rejected.

**Key Questions For Authors:**

I don't understand why the $\mathrm{ERM}_{\mathrm{bit}}$ problem and the process of calculating the bits of the gradient are related. Can you explain in more detail?

**Limitations:**

Yes

**Strengths And Weaknesses:**

Before I begin, I would like to say that I do not have sufficient expertise to evaluate this paper.

Strength

If the content is valid, I think this paper makes a sound discovery for a more practical setting.

Weakness

Overall, I believe this paper is unsuitable for this conference, both in terms of content scope and formatting. This is the first time I've seen a paper without a conclusion and with a prelimilary that starts on page six. If no other reviewer strongly argues that this paper should be accepted, I think it would be right to reject this paper for the sake of conference integrity.

As the authors note, this paper approaches a more practical setting, but it doesn't seem to fully limit the bits of individual calculations, as is done in real floating-point systems. This fact diminishes the paper's value.

---

> ### Author Rebuttal · Authors · 2026-03-30
>
> We deeply thank the reviewer for reading the paper and for the feedback. We also appreciate the reviewer’s candid note that this paper is outside their main area, and we hope the clarifications below address the main concerns.
>
> We respectfully disagree with the claim that the paper is outside the scope of ICML. The conference explicitly includes both Theory of machine learning and Deep learning theory among its topics of interest. Our paper lies exactly at this intersection: it studies the theory of deep network training under a finite-precision bit model, and proves a complexity-theoretic dichotomy between polynomial and piecewise-linear activations.
>
> Regarding presentation, we agree that the structure can be improved, and we appreciate this feedback. In the revision, we will move the discussion to the end of the paper, add a short conclusion, and slightly shorten the introduction so that the technical development begins earlier. That said, the current structure is not unusual for a theory paper. When proofs are long and technical, the main body often focuses on definitions, results, intuition, and proof roadmaps, while substantial proof details are deferred to the appendix. We will revise the organization to make this clearer and more accessible, but we do not believe that this presentation choice makes the paper unsuitable for the conference.
>
> We also want to clarify the reviewer’s concern about practicality. We do not claim that our model is a perfect replica of floating-point hardware. Rather, it is a clean abstraction of finite-precision computation. If one works with exact real numbers, then finite-representation issues disappear entirely, so the question we study cannot even be posed. On the other hand, if one attempts to model every hardware detail exactly, then it becomes difficult to obtain clean and general complexity-theoretic results. Our model is designed to capture the central feature shared by digital computation, namely that numbers are finitely encoded, while remaining mathematically tractable. In this sense, the paper isolates a real finite-precision effect that is invisible in the standard real-RAM framework: for polynomial activations, the difficulty is not merely that intermediate values may become large, but that the relevant bits of these values can themselves encode computationally hard information.
>
> To answer the reviewer’s specific question about why ERM is related to calculating bits of the gradient: in our construction, the gradient is not just an auxiliary quantity that happens to appear during optimization. It is the object that carries the hard arithmetic computation. More concretely, the network is built so that it simulates a straight-line program (an arithmetic circuit with the operations $+,-,\times$) computing a number $n_P$. We then choose a simple loss and a distinguished parameter $w_{e^\star}$ so that, at a specific rational parameter point $\theta^\star$, the corresponding gradient coordinate $\frac{\partial L}{\partial w_{e^\star}}(\theta^\star)$ equals $n_P$ up to a fixed multiplicative factor.
> Thus, the gradient coordinate is directly proportional to the output of the encoded arithmetic circuit. As a result, deciding whether this gradient coordinate is positive is equivalent to deciding whether $n_P>0$, and deciding one of its bits is equivalent to deciding a bit of $n_P$. This is why the bit-complexity of the gradient is directly related to the hardness problem: the reduction explicitly places the hard numerical quantity inside a single gradient entry.
>
> Thus, the issue is not only magnitude, but that the binary representation of the gradient itself can encode hard arithmetic information.

---

> > ### Author Rebuttal · Reviewer_zakj · 2026-04-02
> >
> > The authors' response addressed my concerns well. Still, I lack the expertise to decide the acceptance of this paper. Therefore, I raise my score to weak reject, but the AC may disregard this score. I will not object to this paper being accepted or rejected.

---

### Official Review · Reviewer_eVxk · 2026-03-11

**Soundness:** 2
**Presentation:** 1
**Significance:** 2
**Originality:** 3
**Overall Recommendation:** 3
**Confidence:** 2

**Summary:**

The preprint analyzes Empirical Risk Maximization (ERM) from a theoretical complexity standpoint using a bit-level model that assumes that parameters and inputs are rational rather than real. It proves that ERM for networks with polynomial activation functions is $\sharp\operatorname{P}$-hard, but for piecewise linear functions, it is $\operatorname{NP}$-complete. The submission further suggests that this result offers a complexity-theoretic perspective on the phenomenon of exploding and vanishing gradients in networks with polynomial activation functions.

**Compliance With Llm Reviewing Policy:**

Affirmed.

**Final Justification:**

Overall, I think the paper is well-motivated and proposes an original approach, but the presentation at the time of submission made it hard for me to assess.

The initial submission had many issues, especially around soundness and presentation. The authors have provided an insightful rebuttal that greatly improved my understanding. Proposed changes would also address the concerns I listed. However,  I cannot confidently confirm the improvement without checking a new version. Additionally, because of the large number of issues in the initial submission, I had to focus mostly on major concerns in the initial review, and I cannot be sure which issues will be left in an updated version. Hence, I cannot give more than a Weak Reject at this stage.

I am unfamiliar with the theory related to the $\sharp\operatorname{P}$-completeness and $\exists \mathbb{R}$-completeness that the authors use, so I give my review a low confidence score.

**Key Questions For Authors:**

**Crucial** (Questions affecting the score)

Q1. Presentation: Is it possible to improve the paper structure? Specifically,

 a) Can a discussion or conclusion section be added in the end?

 b) Is it possible to reorganize Section 4 so that it does not end abruptly with the definition?

 c) The Abstract should probably be shortened since it is too long now, contrary to the requirements. (It is 10 sentences, and the requirement is 4-6 sentences).

 d) The Impact Statement is required by the submission guidelines.

 e) I suggest fixing the Appendix that now starts at page 11, which has references.

Q2: Significance: Is there a practical reason why understanding polynomial activations is important?

Q3: Soundness:

a) Can you clarify the connections between the 4 statements in the Introduction and their proofs and justifications appearing later, either in the main part or in the Appendix?

b) What are $\operatorname{PH}$ and $\Delta_2^P$ in Corollary 1.2?

c) What is the proof of Theorem 1.4?

d) Is it possible to design meaningful experiments confirming the theoretical results, or why are only theoretical results sufficient?

e) Can you provide an example where exploding or vanishing gradients would occur due to the discovered theoretical complexity results?

**Minor** (Questions to confirm my understanding and minor errors)

Q4. What exactly is the relation between $\sharp\operatorname{P}$-hardness and $\exists \mathbb{R}$-hardness?

**Limitations:**

yes

**Strengths And Weaknesses:**

**Strengths**

S1. The submission proposes an original approach to the complexity-theoretical analysis of the general ERM that highlights the benefits of piecewise linear activation functions over polynomial activations, explains the importance of finite-precision constraints for learnability, and offers a new perspective on vanishing and exploding gradients.

**Weaknesses**

W1. The presentation in the submission is confusing, and the preprint reads unfinished. The paper abruptly ends with a definition, and there is no conclusion, while the Discussion section appears in the Introduction. See further details in the Key Questions For Authors. Overall, I struggled to assess other aspects of the submission because of the presentation issues.

W2. The significance of the results is not fully clear, as the paper focuses on comparing polynomial activation functions with piecewise linear ones, such as ReLU, but the former are not used in practice.

W3. The soundness of the results is hard to verify. The Abstract and Introduction discuss activation function comparison, but it is hard to see how the formal language in Sections 2, 3, and 4 maps onto the simplified explanations in the Introduction. For instance, I was unable to find formal results in the later sections in the main part discussing piecewise linear activation functions; there is no proof of Theorem 1.4 in the preprint; and some notation is not defined. Additionally, there are no experiments confirming the results.

*Kindly note that it is hard for me to judge the soundness of the submission since I am unfamiliar with the theory related to the $\sharp\operatorname{P}$-completeness and $\exists \mathbb{R}$-completeness.*

---

> ### Author Rebuttal · Authors · 2026-03-30
>
> We deeply thank the reviewer for reading our paper and for the constructive feedback. We take the comments seriously and will revise the paper accordingly. In particular, we will improve the presentation, readability, and structure by adding a clearer proof roadmap, shortening the abstract, smoothing the end of Section 4, moving the discussion to the end of the paper, adding a conclusion section, and clarifying more explicitly the significance of the results and the role of polynomial activations.
>
>
> We answer the reviewer’s questions below and hope these clarifications address the main concerns.
>
>
> Q1.
>
> a) Yes. We will add both a discussion and a conclusion at the end of the main body. We agree that the current placement of the discussion is not ideal, and in the revision, we will move it to a more standard location.
>
> b) Yes. We agree that Section 4 currently ends too abruptly. In the revision, we will smooth the end of Section 4 by adding a short closing paragraph with more intuition for the proof. We will also defer Definition 4.2 to the appendix.
>
> c) We agree that the abstract is too long relative to the 4 - 6 sentence guideline. In the revision, we will shorten it substantially and remove technical detail, keeping only the conceptual message and only the main results.
>
> d) We agree and will restore the accidentally omitted impact statement (in the standard version).
>
> e) We will start the appendix on a new page.
>
>
> Q2. We agree that polynomial activations are not the dominant choice in modern deep learning. Nevertheless, they are closely related to polynomial-network models that have been studied in practical prediction settings, and they also arise naturally as polynomial or rational approximations of smooth activations in finite-precision implementations (see references within our paper, in the second discussion paragraph). They therefore provide a clean contrast with piecewise-linear activations such as ReLU. Our result is precisely that this algebraic distinction leads to a sharp complexity separation in the bit model, and we will state this motivation more clearly in the revision.
>
>
> Q3: Soundness.
>
> a) Yes. We agree that the introduction should point more explicitly to where the main statements are proved. In the revision, we will add an explicit proof roadmap indicating where the multiplication-gadget reduction for Theorem 1.1  and Corollary 1.2, the PosSLP/BitSLP reductions for Theorem 1.3, and the piecewise-linear forward/backward bounds underlying Theorem 1.4 are established (Appendix E). We believe this will make the logical structure of the paper much clearer.
>
> b) We will add a brief explanation that $PH$ is the polynomial hierarchy and that its second level is $\Delta_2^P = P^{NP}$, the class of problems solvable in polynomial time with an $NP$ oracle. Thus, Corollary 1.2 says that if $ERM_{bit} \in NP$, then the polynomial hierarchy collapses to its second level.
>
> c) The proof of Theorem 1.4 is given in Appendix E. Specifically, Theorem 1.4 follows from Theorem E.2 together with the NP-hardness argument from prior work. The proof has two main parts: first, for piecewise-linear activations, exact forward computation and exact backpropagation preserve polynomially bounded rational bit complexity, since each node performs only comparisons and affine operations with rational coefficients; second, NP-hardness follows from existing hardness results for training ReLU networks, since ReLU is a special case of piecewise-linear activations. We will make this proof route explicit in the theorem statement and proof roadmap.
>
> d) Since our main results are complexity-theoretic hardness statements, experiments can only be illustrative, not evidentiary. In particular, they cannot establish #$P$-hardness or the conditional intractability of deciding the sign of a gradient coordinate. Still, we agree that illustrations are useful, and in the revision, we will extend Figure 1 with additional examples comparing polynomial and piecewise-linear activations across depth, highlighting the rapid bit-growth in the former and the much milder growth in the latter.
>
> e) Yes: already in a depth-$L$ network with $\sigma(x)=x^2$, repeated squaring makes gradients blow up from $x=2$ and vanish from $x=\tfrac12$. Practical heuristics such as clipping, normalization, and careful initialization mitigate this in many cases, but they do not remove the worst-case phenomenon captured by our results.
>
> Q4. $\exists\mathbb{R}$-hardness and #$P$-hardness capture different sources of intractability: the former concerns the existence of real solutions to polynomial inequalities, while the latter concerns counting and bit-computation difficulty. Our point is not that one implies the other, but that moving from the classical real-valued model to the bit model changes the type of hardness that governs training, and we will clarify this distinction in the revision.

---

> > ### Author Rebuttal · Reviewer_eVxk · 2026-04-03
> >
> > I thank the authors for their detailed responses to my questions. This has improved my understanding of the contribution and confidence in the soundness of the results. I also appreciate the changes the authors propose to improve the submission, which would address the concerns I outlined.
> >
> >
> > Unfortunately, given the number of initial concerns I had, especially around soundness and presentation, I cannot properly judge the improvement without checking the updated version, which is impossible under the current system. I will update my score from Reject to Weak Reject to reflect the rebuttal. However, as stated in the review, I am not knowledgeable about the used theory, so my review has a low confidence score.

---

### Official Review · Reviewer_VxYq · 2026-03-13

**Soundness:** 4
**Presentation:** 2
**Significance:** 3
**Originality:** 4
**Overall Recommendation:** 4
**Confidence:** 3

**Summary:**

This work analyzes the theoretical complexity of computing ERM under a bit-complexity model with polynomially-bounded bit lengths. It is a refinement over prior work analyzing the complexity when using real numbers, as it is closer to realistic settings when these computations are done on a finite-sized hardware. The results are extended to analyze gradient computations as well. Intererstingly, the results require the use of non-linear polynomial activation functions, and in particular exclude the use of ReLU.

**Compliance With Llm Reviewing Policy:**

Affirmed.

**Final Justification:**

I remain unconvinced that there are practical applications. The exponential growth in the number of layers depending on the activation function doesn't require a bit-specific analysis. Nevertheless, the theoretical contributions are interesting, so I will maintain my rating.

**Key Questions For Authors:**

1. To check my understanding, is the bit complexity of intermediate states of the model allowed to grow, even though the bit complexity of the input and parameters is restricted?

2. Is the growth in the bit complexity as you get deeper into the network the reason why ERM becomes hard to compute?

3. If the complexity results are due in part to the bit complexity of intermediate results growing at a large rate, does this model still capture the "reality of digital computation with finite-precision hardware"?

**Limitations:**

yes

**Strengths And Weaknesses:**

This work provides a novel analysis of the complexity of computing ERM, in particular making use of a multiplication gadget which is available when non-linear polynomial activation functions are used. The use of a finite-precision model over a one using real numbers is well motivated. However, some of the important details of the model could be made more explicit. In addition, the relevance to realistic models is overstated.

---

> ### Author Rebuttal · Authors · 2026-03-30
>
> We deeply thank the reviewer for reading our paper and for their helpful comments and questions. We take the reviewer’s feedback very seriously and plan to revise the paper accordingly. In particular, we will give a more explicit definition of the model along with necessary clarifications, including those that arise from the reviewer's questions regarding intermediate states (see our answers below). In addition, we will clarify more precisely the limitations of the proposed theoretical model of ERM to capture exactly practical settings.
>
> We answer the reviewer’s questions below.
>
>
> Question: To check my understanding, is the bit complexity of intermediate states of the model allowed to grow, even though the bit complexity of the input and parameters is restricted?
>
> Answer: Yes. In our formulation, the polynomial bit-length bound is imposed on the input instance and any parameter vector (solution) $\theta$, which are the objects explicitly represented in the problem statement. Intermediate quantities are then computed exactly by rational arithmetic in the standard bit model, so their encoding length may grow during evaluation. We view this as a natural modeling choice: the input and weights are required to be succinctly representable, while the complexity question is precisely whether computation from such succinct data remains polynomially manageable. Our results show a sharp dichotomy here: for piecewise-linear activations, this precision growth remains polynomially controlled, whereas for nonlinear polynomial activations, it can become intractable, even when the magnitudes of all intermediate values are uniformly bounded.
>
> Question: Is the growth in the bit complexity as you get deeper into the network the reason why ERM becomes hard to compute?
>
> Answer: Essentially yes, the growth in the precision/bit complexity of intermediate computations, not merely their numerical magnitude, is what underlies the hardness. For nonlinear polynomial activations, increasing depth allows the network to compose arithmetic in a way that can encode hard deep arithmetic circuits, making exact evaluation of the ERM objective intractable in the bit model. In contrast, for piecewise-linear activations, this precision growth remains polynomially controlled, which is why $ERM_{bit}$ stays in NP in that case. Importantly, our hardness still holds even when all intermediate values are uniformly bounded in magnitude, so the issue is not simply that the numbers become large.
>
> Question: If the complexity results are due in part to the bit complexity of intermediate results growing at a large rate, does this model still capture the "reality of digital computation with finite-precision hardware"?
>
> Answer: We agree that our model does not exactly match modern hardware in practice: it does not capture fixed-width floating-point formats, per-operation rounding, etc. Still, we believe it is a natural and useful complexity-theoretic abstraction of digital computation, since it requires inputs and parameters to be finitely encoded bit strings rather than arbitrary real numbers, which has been the canonical ERM formulation in deep learning theory. Some abstraction is necessary to obtain meaningful theoretical results: for example, if one models practical numbers only as fixed-width machine words, then each arithmetic operation is O(1) by definition, thereby hiding the precision-growth phenomenon we aim to study. Our question is whether computation from such succinct digital data remains tractable as depth increases. Our results show that, for nonlinear polynomial activations, the required intermediate precision can become computationally intractable, whereas for piecewise-linear activations it remains polynomially controlled. We will clarify this positioning and limitation explicitly in the paper.

---

> > ### Author Rebuttal · Reviewer_VxYq · 2026-04-04
> >
> > Thank you for the clarifications, they significantly help with understanding the intuition, and will make the presentation clearer. The theoretical contributions are nice, although it is still unclear to me whether or not there is any significance to realistic models (for instance, whether this says anything about whether to use ReLU or other polynomial nonlinear activation functions). Question: is there?
> >
> > For those reasons, I will maintain my positive rating.

---

> > > ### Author Response · Authors · 2026-04-04
> > >
> > > Thank you for the response and your feedback. We will add the intuition from the rebuttal to the paper. Our paper gives a dichotomy between the two activation types (polynomials and piecewise-linear functions; for the following discussion, let us fix $x^2$ and ReLU as canonical examples of these classes, respectively) using complexity-theoretic machinery. Specifically, in the bit model, ERM and backpropagation are much more tractable for piecewise-linear activations than for non-linear polynomial activations, intuitively because intermediate precision remains better controlled. While this bit model is naturally an abstraction of practical deep learning, it still suggests a meaningful dichotomy between these activation types for core optimization tasks in a bit model much closer to practice, compared to real-RAM.
> > >
> > > Furthermore, even without the time-complexity lens, in the bit model, the advantages of ReLU over $x^2$ in terms of bit growth across layers can also be seen directly. Consider, as a toy example, a network of width $1$ and $L$ layers; then, for ReLU the magnitude at layer $L$ grows at most exponentially in $L$ (for instance, on the order of $(|x| \cdot W_{\max})^{L}$, where $W_{\max}$ is the largest weight magnitude in the network and $x$ is the input). In sharp contrast, for non-linear polynomials such as $x^2$, the magnitude at layer $L$ can be as large as $(|x| \cdot W_{\max})^{2^L}$, which is astonishingly larger for deep networks. While there are practical ways to clip such overflowing values, these networks are much less stable and require more aggressive clipping, which can have negative downstream effects and make these activations harder to use robustly in practical deep networks.
> > >
> > > Hence, we believe our complexity-theoretic results provide formal support for an effect that is easily observable in practice and may also be of interest to practitioners. We agree with the reviewer that clarifying this point will increase the practical relevance of our work, and we will revise the paper accordingly.

---

### Official Review · Reviewer_eLy7 · 2026-03-15

**Soundness:** 3
**Presentation:** 3
**Significance:** 3
**Originality:** 3
**Overall Recommendation:** 4
**Confidence:** 3

**Summary:**

The paper studies neural network training under a finite-precision bit model rather than the usual real-valued ERM formulation. Its main claim is a dichotomy by activation type: for deep networks with non-linear polynomial activations, training in the bit model becomes very hard (#P-hard with regularization), and even computing gradient information is intractable in a strong sense; for piecewise-linear activations such as ReLU, the corresponding bit-model training problem is NP-complete and exact backpropagation is polynomial-time. The paper argues that this helps explain why piecewise-linear activations are especially favorable in deep learning.

**Compliance With Llm Reviewing Policy:**

Affirmed.

**Key Questions For Authors:**

How essential is the centred regulariser  to the P-hardness result? Do you expect the same qualitative hardness for the plain ERM objective or standard weight decay?

The backpropagation sign hardness is conditional on the constructive univariate radical conjecture. How should readers think about the strength of the result if they do not want to rely on that conjecture?

For piecewise-linear activations, the paper proves NP-completeness of ERMbit and polynomial-time exact backprop. What is the most important conceptual reason that piecewise-linearity prevents the bit-growth phenomenon used in the hard cases?

**Strengths And Weaknesses:**

Short strengths
Clear and interesting theoretical question: replacing the idealised real-RAM view with a finite-precision bit model is well motivated.
The polynomial-vs-ReLU dichotomy is sharp and easy to understand at a high level.
The results go beyond ERM hardness and also cover gradient computation, which makes the story more compelling.
The discussion section is thoughtful about scope, limitations, and how the model relates to practical quantised training.

Short weaknesses
Some headline hardness results rely on a non-standard loss and a centred quadratic regulariser, which weakens the practical force of the conclusions.
Part of the backpropagation hardness is conditional, relying on an additional conjecture.
The positive result for ReLU is mostly a verification/backprop tractability statement in the bit model, not a claim that training is easy in practice.
The paper leaves important activation families unclassified, including common smooth nonlinearities except through fixed finite-precision approximations.

---

> ### Author Rebuttal · Authors · 2026-03-30
>
> We deeply thank the reviewer for their thorough reading of our paper and for their helpful comments and questions. We answer their questions below.
>
>
> Question 1:
>
> We agree that the centered regularizer can be viewed as a limitation of the current formulation of Theorem 1.1. However, as we mentioned briefly in the discussion, we have verified that the same qualitative hardness presented in Theorem 1.1 can also be obtained for the ERM objective without a regularizer, at the cost of increasing the input and output dimensions polynomially. The main idea is to keep the same SLP-simulating network with the same distinguished parameter vector, and the same final bit-extraction step, while exposing intermediate node values as additional input/output coordinates and adding polynomially many auxiliary samples that enforce each edge weight must choose the distinguished parameter vector to have a small loss. Specifically, for identity edges, a few such samples force the correct linear map, since a linear function is uniquely determined by its values on sufficiently many chosen inputs.
> For a polynomial activation edge with a polynomial of degree $\mu$, $\mu+1$ carefully chosen samples force the correct map, since a degree-$\mu$ univariate polynomial is uniquely determined by its values on $\mu+1$ distinct inputs.
> Repeating these auxiliary samples sufficiently many times ensures that any deviation from the distinguished parameter vector already exceeds the allowed loss threshold. Thus, the rest of the reduction remains unchanged, and only the mechanism used to anchor the correct computation is modified.
> This point concerns only Theorem 1.1. Our current proof uses the centered regularizer because it gives the cleanest bounded-dimension formulation. The unregularized extension is technically more involved, but we have carefully verified its correctness. We do not currently claim the same result with constant input and output dimensions apply for standard weight decay, and we will clarify this distinction and include the unregularized extension in the revised version. We will be happy to share more details if useful.
>
>
> Question 2:
>
> Indeed, formally, the result is stated conditioned on this conjecture. Practically, the hardness stems from the problem of determining the sign of an SLP (PosSLP). As the complexity of PosSLP is not fully understood, and its most canonical hardness statement is conditional on the constructive univariate radical conjecture, we state the theorem this way. However, PosSLP is widely believed to be hard (in particular, not in P), and resolving its complexity is a central question in arithmetic circuit complexity. Thus, expressing hardness in terms of PosSLP is already meaningful. In particular, our reduction shows unconditionally that a polynomial-time algorithm for BackpropSign would imply a polynomial-time algorithm for PosSLP; the conjecture is only needed to derive the stated hardness classification. More broadly, conditional complexity results are standard and often the only available way to assess hardness, with the most famous example being the assumption $P \neq NP$. Following the reviewer's comment, we will revise the paper to emphasize that the hardness of BackpropSign can also be stated directly in terms of the hardness of PosSLP. We also note that we show the related problem BackpropBit is \# $P$-hard unconditionally, so the conjecture is only needed for determining the sign of the gradient, while the hardness of determining a specific bit in the gradient admits a stronger hardness result.
>
>
> Question 3:
>
> Intuitively, the reason that piecewise (pw)-linear activations do not cause bit-growth is that applying such an activation to an input $x$ can increase the bit-length of the output by at most an additive constant. For example, applying the pw-linear function $f(x)=2\operatorname{ReLU}(x)$ to any rational input $x$ may increase, in standard binary encoding, the bit-length of $x$ by at most $1$, corresponding to a single left shift. As we explain in Appendix E, the same holds for any pw-linear function. In contrast, non-pw-linear functions such as $f(x)=x^2$ generally lead to a constant multiplicative increase in bit-length. For example, if $x$ has binary bit-length $k$, then $x^2$ has binary bit-length about $2k$ (up to an additive constant). We will adjust the paper to explain this more clearly.

---

### Decision · Program_Chairs · 2026-04-30

**Decision:**

Accept (regular)

**Comment:**

The paper studies neural network training in a finite-precision bit model and establishes a sharp complexity-theoretic dichotomy between polynomial and piecewise-linear activations.

Reviewers generally agree that the question is original, well motivated, and technically interesting, and the rebuttal successfully clarified several concerns, especially regarding the model, the role of bit growth, and the interpretation of the main hardness results.

Some weaknesses remain, mainly around presentation and the practical scope of the conclusions, but these seem addressable in revision.